# Crustal heat production and estimate of terrestrial heat flow in central East Antarctica, with implications for thermal input to the East Antarctic ice sheet

John W. Goodge[1]

[1]Department of Earth and Environmental Sciences, University of Minnesota, Duluth, MN 55812 USA

*Correspondence to*: John Goodge (jgoodge@d.umn.edu)

**Abstract.** Terrestrial heat flow is a critical first-order factor governing the thermal condition and, therefore, mechanical stability of Antarctic ice sheets, yet heat flow across Antarctica is poorly known. Previous estimates of terrestrial heat flow in East Antarctica come from inversion of seismic and magnetic geophysical data, by modeling temperature profiles in ice boreholes, and by calculation from heat production values reported for exposed bedrock. Although accurate estimates of

surface heat flow are important as an input parameter for ice-sheet growth and stability models, there are no direct measurements of terrestrial heat flow in East Antarctica coupled to either subglacial sediment or bedrock. As has been done with bedrock exposed along coastal margins and in rare inland outcrops, valuable estimates of heat flow in central East Antarctica can be extrapolated from heat production determined by the geochemical composition of glacial rock clasts eroded from the continental interior. In this study, U, Th and K concentrations in a suite of Proterozoic (1.2-2.0 Ga)

granitoids sourced within the Byrd and Nimrod glacial drainages of central East Antarctica indicate average upper crustal heat production ($H_o$) of about $2.6 \pm 1.9$ $\mu$W m$^{-3}$. Assuming typical mantle and lower crustal heat flux for stable continental shields, and a length scale for the distribution of heat production in the upper crust, the heat production values determined for individual samples yield estimates of surface heat flow ($q_o$) ranging from 33-84 mW m$^{-2}$ and an average of $48.0 \pm 13.6$ mW m$^{-2}$. Estimates of heat production obtained for this suite of glacially-sourced granitoids therefore indicate that the interior of

the East Antarctic ice sheet is underlain in part by Proterozoic continental lithosphere with average surface heat flow, providing constraints on both geodynamic history and ice-sheet stability. The ages and geothermal characteristics of the granites indicate that crust in central East Antarctica resembles that in the Proterozoic Arunta and Tenant Creek inliers of Australia, but is dissimilar to other areas like the Central Australian Heat Flow Province that are characterized by anomalously high heat flow. Age variation within the sample suite indicates that central East Antarctic lithosphere is

heterogeneous, yet the average heat production and heat flow of four age subgroups cluster around the group mean, indicating minor variation in thermal contribution to the overlying ice sheet from upper crustal heat production. Despite these minor differences, ice-sheet models may favour a geologically realistic input of crustal heat flow represented by such a distribution of ages and geothermal characteristics.

# 1 Introduction

Heat production and heat flow are fundamental characteristics of continental crust (Rudnick and Fountain, 1995). Together they provide important constraints on the thermal and petrogenetic history of cratonic lithosphere, and heat flow is an indicator of modern geodynamic environments. Antarctic lithosphere is uniquely important because it underlies Earth's largest ice caps, including numerous subglacial lakes, and it is critical in governing the thermal state and mechanical stability of overlying ice (Pollard et al., 2005; Jamieson and Sugden, 2008; Van Liefferinge and Pattyn, 2013; Schroeder et al., 2014). Terrestrial heat flow in Antarctica has a strong influence on basal ice temperatures, amount of basal ice at its pressure melting point, and the formation of liquid water, all of which affect basal ice conditions, mechanical properties of glacial bed material, degree of basal sliding, erosional effectiveness, and the distribution of hydrologic networks and subglacial lakes (e.g., Siegert, 2000; Pollard et al., 2005; Pollard and DeConto, 2009). Despite its importance in governing ice-sheet mass balance—and therefore as an input parameter for ice-sheet growth and stability models—only a few estimates of conductive heat flow are available from measurement in subglacial sediment or from temperature profiles in Antarctic ice (e.g., Begeman et al., 2017; Fischer et al., 2013). This is particularly problematic for East Antarctica, where the ice cap exceeds 4 km in many areas. In order to develop accurate models of past ice-sheet behavior and forward models of ice-sheet stability, it is therefore crucial to have good estimates of terrestrial heat flow from East Antarctica.

Continent-wide models for terrestrial heat flow come from both seismological and satellite magnetic data. To address a lack of direct heat flow measurements in Antarctica, Shapiro and Ritzwoller (2004) modeled surface heat flow by first correlating seismic velocity data from crust and upper mantle in regions of known heat flow, and then extrapolating these results to a seismic model of Antarctic lithosphere. Over a broad region of East Antarctica they estimated surface heat flow to be notably low, uniform, and similar to other old cratons (mostly 35-60 mW m$^{-2}$, with a mean estimate for East Antarctica of 57 mW m$^{-2}$). Similarly, An et al. (2015) used a 3-D S-wave velocity model to construct temperature profiles for Antarctic lithosphere, from which they derived an average surface heat flux of 47 mW m$^{-2}$ for the central Gamburtsev Subglacial Mountains region in East Antarctica. Fox Maule et al. (2005) used satellite magnetic data to model the depth to Curie temperature, and then inverted the resulting thermal profile to generate a distribution of heat flow (see also Purucker, 2012); this modeling likewise predicted heat flow in East Antarctica to be similar to the results obtained from seismology, with average heat flux of 50-60 mW m$^{-2}$ in the central part of East Antarctica. In order to evaluate areas that may preserve very old ice, generally requiring relatively thick, slow-moving ice under cold conditions with low basal heat flux, Van Liefferinge and Pattyn (2013) derived an average distribution of heat flow from a simple mean of existing geophysical models (Fig. 1); this continent-wide synopsis highlights a relatively uniform pattern of low heat flow in East Antarctica (mostly <55 mW m$^{-2}$). Using a thermal model that assumes basal ice temperatures above Antarctic subglacial lakes are equal to the pressure-melting value, Siegert (2000) estimated geothermal heat flow to vary between 37 and 65 mW m$^{-2}$, although most estimates for East Antarctica are <60 mW m$^{-2}$. In general terms, these different models, despite coarse kernel size, are consistent with one another and indicate heat flow in most of East Antarctica between about 35-60 mW m$^{-2}$.

In addition to models based on remote geophysical observations, there are also some field-based estimates. Using a measured temperature profile in the EPICA ice borehole at Dome C, Fischer et al. (2013) derived a geothermal heat flux of about 54 mW m$^{-2}$ by fitting a model of heat flow and basal ice melting to the thermal profile. A geological approach was taken by Carson et al. (2014), who derived heat flow using values for heat production estimated from the abundances of radioactive elements in crustal rocks sampled from outcrop near the Amery Ice Shelf (compiled by Carson and Pittard, 2012). From a coastal transect across rock exposures at Prydz Bay, the resulting profile indicates that heat flow in this area of Archean to Proterozoic igneous and metamorphic crust is highly variable over a distance of about 200 km, ranging from an average of 31 mW m$^{-2}$ in the Vestfold Hills, to 44 mW m$^{-2}$ in the Rauer Islands, and 55-70 mW m$^{-2}$ in the area of southern Prydz Bay. Locally, high heat-producing Cambrian granitoids indicate heat flow values as high as ~85 mW m$^{-2}$. Their model results thus show variable heat flow governed to first order by the age and type of crust represented, and punctuated by heat production spikes contributed from Th-rich granitoids. Although much of the area underlying the Rauer Islands and Vestfold Hills have low heat flow (<50 mW m$^{-2}$) typical of Proterozoic and Archean crust, Carson et al. (2014) emphasized that some early Paleozoic granites with anomalously high heat production can cause local elevation of heat flow (>80 mW m$^{-2}$), as observed in the Central Australian Heat Flow Province (McLaren et al., 2003). Thus, spatially coarse models of heat flow based on geophysical data across East Antarctica indicate relatively typical continental values ranging from about 35-60 mW m$^{-2}$, yet there are some indications of locally elevated heat flow in areas underlain by high heat-producing granites.

Although there are no measurements of terrestrial heat flow obtained directly from subglacial sediment or rock in East Antarctica, rock clasts eroded from the continental interior and transported to the margin can provide insight into the subglacial geology and, therefore, heat production. In this paper, the concentrations of heat-producing elements in glacial igneous rock clasts provide a unique opportunity to assess heat flow in the deep continental interior. The major ice drainages in East Antarctica are marked by nearly radial flow away from central ice divides and domes toward the continental margin (Fig. 2), providing natural proxy samples of the continental interior by bedrock erosion during glacial flow (Peucat et al., 2002; Goodge et al., 2008, 2010, 2017). Unique among the major drainages, glacial ice in Byrd Glacier and related smaller drainages moves non-radially from the main ice divides because it is obstructed by the high-standing Transantarctic Mountains (peak elevations >4000 m). Ice flows through the mountains via channelized outlet glaciers, but it also ablates in areas where it ramps up against the mountain range (Whillans and Cassidy, 1983). Glacial moraines are formed both along the margins of the outlet glaciers and where ice is ablating, forming lag deposits adjacent to the mountains. As part of a study of East Antarctic crustal history, glacial moraines were sampled at sites between the Byrd and Beardmore glaciers (Fig. 2). Ice-velocity fields show that material transported in the greater Byrd system may have been eroded from a broad area of central East Antarctica, potentially from near the upstream boundary along the major ice divide connecting Dome A and Dome C.

As proxies for subglacial geology, igneous clasts eroded from central East Antarctica and collected from moraines adjacent to the central Transantarctic Mountains were dated and analyzed geochemically for major and trace elements—including the major heat-producing elements U, Th and K—yielding values of radiogenic heat production. Surface heat flow

was then estimated by assuming mantle and lower crustal heat-flux contributions and a length scale for reduction in upper-crustal radiogenic heat production. The results indicate that heat production among most of the samples in this group is within the range expected for average continental crust and that terrestrial heat flow for a large region of central East Antarctica is like that commonly observed in Precambrian shield areas. Estimates from this analysis corroborate previous
geophysical models of heat flow in East Antarctica and can be used as first-order constraints in ice-sheet and lithospheric thermal models.

## 2  Glacial igneous clasts

Glacial igneous rock clasts were collected as part of a separate study to obtain lithologic, petrologic, and isotopic information about the Precambrian crust of East Antarctica (Goodge et al., 2017). About a dozen sites were sampled where
glacial moraines are exposed, extending >1500 km along the length of the Transantarctic Mountains from the Convoy Range in southern Victoria Land to Strickland Nunatak near Reedy Glacier (Fig. 2). The original purpose of the clast sampling was to obtain a representative set of samples from the ice-covered East Antarctic craton in order to address questions of crustal age, composition, and Precambrian history by investigating their age and isotopic compositions. In the field, any rock clasts that represented potential cratonic basement were collected within allotted ground time, thus providing an effectively
randomized sample. Over 300 large individual clasts were obtained. For the previous study, five sites yielded the most useful samples; the other sites were dominated by clasts of Beacon Supergroup sediment or Ferrar dolerite eroded from the Transantarctic Mountains. A major effort was then undertaken to screen the crystalline clast samples by reconnaissance *in situ* U-Pb geochronology in order to cull samples with Ross Orogen (~500 Ma) or younger ages, in order to focus solely on the Precambrian crustal history of the shield interior. Analysis of the remaining suite of 22 samples included detailed
petrography, mineral analysis, geochemical analysis, mineral separation, precise U-Pb geochronology, O-isotope analysis, and Hf-isotope analysis (see Goodge et al., 2017).

In this contribution, results are presented from samples collected at three sites for which geochemical data are available from dated Precambrian rock clasts—Lonewolf Nunataks (LWA and LWB), Mt. Sirius (MSA), and Turret Nunatak (TNA) (Fig. 2; Table 1). At Lonewolf Nunataks, elongate bands of distributed moraine and ice-matrix debris follow narrow flow
lines related to ice movement along the southern margin of Byrd Glacier. This site, at the southern margin of Byrd Glacier, was the single most productive site sampled, presumably because the Byrd ice stream is among the fastest outlet glaciers traversing the Transantarctic Mountains and capable of significant glacial erosion. It is numerically over-sampled compared to other sites (Table 1), yet it contains the full range of clast ages obtained for the whole suite (1.2-2.0 Ga) so is likely to be representative of the craton interior. Sites at Turret Nunatak and Mt. Sirius are dominated by Gondwanide debris, but they
also yielded a small number of distinctive crystalline clasts. Together, these sites comprise a suite of pre-Ross igneous samples with ages ranging from about 1.2-2.0 Ga. The igneous clasts consist mainly of intermediate to felsic igneous rocks that represent magmatic components of the ice-covered East Antarctic craton. They are granitic to granodioritic in

composition and contain hornblende, biotite and/or muscovite; some samples are two-mica granitoids of peraluminous composition. Detailed sample descriptions are provided by Goodge et al. (2017), including petrographic features, geochemical data, cathodoluminescence (CL) images of zircons, and zircon isotopic data (U-Pb, O, and Lu-Hf).

Zircon U-Pb ages from this suite of glacially-transported granitoid clasts show that the crust in central East Antarctica was formed by a series of magmatic events at ~2.01, 1.88-1.85, ~1.79, ~1.57, 1.50-1.41, and 1.20-1.06 Ga (Goodge et al., 2008, 2010, 2012, 2017). The dominant granitoid populations are ca. 1.85, 1.45 and 1.20-1.06 Ga. None of these igneous ages are known from the limited outcrop in the region. Samples of metamorphic rock clasts from the same moraines have similar Proterozoic ages ranging from about 1.1-1.9 Ga (Goodge et al., 2010; Nissen et al., 2013). By comparison with nearby mountain outcrops, the types and ages of these samples indicate that the crust of central East Antarctica comprises plutonic and metamorphic rocks unlike those seen in the central Transantarctic Mountains (Nimrod Complex ages of ca. 3.1, 2.5, and 1.7 Ga; Goodge et al., 2001; Goodge and Fanning, 2016). Likewise, they are different from igneous and metamorphic rocks exposed at the Terre Adélie coast (ages of ca. 2.4 and 1.7 Ga; Oliver and Fanning, 1997), although one population is similar in age to ~1.6 Ga glacial clasts sampled by Peucat et al. (2002). As shown in Figure 2, the glacially-eroded igneous clasts discussed here may also sample the Gamburtsev Subglacial Mountains, which is thought to have nucleated growth of the East Antarctic Ice Sheet (DeConto and Pollard, 2003; Bo et al., 2009; Rose et al., 2013).

It is important to note that only with high-quality age data for this igneous clast suite is it possible to constrain potential heat production and heat flow within the west-central interior of East Antarctica. In the absence of age data, the origin of glacially-transported clasts is largely unconstrained and a large fraction of any such samples could be sourced from the Ross Orogen or from younger Beacon cover in the Transantarctic Mountains, neither of which inform subglacial heat flow in the craton interior. Thus, simply sampling moraines to obtain a large set of geochemical data without geochronological control may yield misleading results. Although it would be beneficial to have a larger sample set taken from a potentially wider catchment area, this would require substantial logistical and analytical resources beyond those employed by this initial reconnaissance study.

## 3 Analytical methods

Bulk-rock X-ray fluorescence (XRF) and inductively-coupled plasma mass spectrometry (ICPMS) analyses of major and trace element compositions were completed in the GeoAnalytical Lab at Washington State University (Goodge et al., 2017). Prior to analysis, fresh chips of each sample were hand-picked and a standard amount (approximately 28 g) was ground in a swing mill with tungsten carbide surfaces for 2 minutes. For XRF analysis of major elements, 3.5 g of sample powder was weighed into a plastic mixing jar with 7 g of spec pure dilithium tetraborate ($Li_2B_4O_7$). The mixed powders were emptied into graphite crucibles and loaded into a muffle furnace for fusion at 1000 °C. After removing from the oven to cool, each bead was reground in the swing mill and the resulting glass powders were replaced in the graphite crucibles and refused for 5 minutes, then cooled to form a glass bead. Their lower flat surfaces were then ground on 600 silicon carbide

grit and finished briefly on a glass plate to remove any metal from the grinding wheel. The concentrations of 29 elements in the unknown samples were measured on a ThermoARL Advant'XP+ sequential X-ray fluorescence spectrometer by comparing the X-ray intensity for each element with the intensity obtained from USGS standard samples (PCC-1, BCR-1, BIR-1, DNC-1, W-2, AGV-1, GSP-1, G-2, and STM-1, using the values recommended by Govindaraju (1994) and beads of pure vein quartz used as blanks for all elements except Si. Twenty standard beads are routinely run and used for recalibration approximately once every three weeks or after the analysis of about 300 unknowns. The intensities for all elements were corrected for line interference and absorption effects due to all the other elements using the fundamental parameter method.

For trace elements, powdered samples were mixed with 2 g of $Li_2B_4O_7$ flux, placed in a carbon crucible and fused at 1000 °C in a muffle furnace for 30 minutes. After cooling, the resultant fusion bead was briefly ground in a carbon-steel ring mill and a 250 mg portion was weighed into a 30 ml, screw-top Teflon PFA vial for dissolution in water, $HNO_3$, $H_2O_2$, and HF and warmed on a hot plate until a clear solution was obtained. Samples were then diluted to a final weight of 60 g with de-ionized water. Solutions were analyzed for 27 elements on an Agilent model 4500 ICPMS and were diluted an additional 10x at the time of analysis using Agilent's Integrated Sample Introduction System (ISIS). This yielded a final dilution factor of 1:4800 relative to the amount of sample fused. Instrumental drift was corrected using Ru, In, and Re as internal standards, and applying a linear interpolation between In and Re to compensate for mass-dependent differences in the rate and degree of instrumental drift. Isobaric interferences of rare-earth and other oxides were optimized with correction factors using mixed-element solutions. Standardization was accomplished by analyzing duplicates of three in-house rock standards interspersed within each batch of 18 unknowns.

## 4  Heat production and estimated heat flow

### 4.1  Sample geochemical characteristics

Major, trace and rare-earth element geochemical data show that the granitoid samples are Si-rich with >65 wt% $SiO_2$, and many have $SiO_2$ = 70-75 wt%. Trace-element abundances are enriched with light rare-earth elements (LREE) and depleted in heavy rare-earth elements (HREE) relative to chondrites, and they are enriched in large ion lithophile (LIL) elements and slightly depleted in high field-strength (HFS) elements relative to mid-ocean ridge basalt (MORB). Their trace and rare-earth element signatures are quite similar to modern continental-margin magmatic arc systems (e.g., Cascades, Andes) or evolved volcanic arcs, and they show very similar patterns and abundances as magmas interacting with thick crust (e.g., Davidson et al., 1990, 1991; Wörner et al., 1994). Some of the samples resemble Si-rich, peraluminous leucogranites found in regions of over-thickened continental crust (Frost et al., 2001). In broad terms, then, the trace-element compositions indicate that the melts that produced these igneous rocks interacted with thick, evolved continental crust, but that they are dissimilar generally from intraplate granitoids.

### 4.2  Heat-producing elements

The concentrations of the heat-producing elements U, Th and K in 18 granitoid samples are listed in Table 1. The concentration of U is generally low, ranging up to about 6 ppm (mean = 2.0). Thorium ranges widely, from 1-98 ppm (mean = 23.7), and K similarly varies from 0.5-8 wt% $K_2O$ (mean = 4.34). Most of the samples have normal concentrations of U, Th and K quite similar to the ranges expected for Proterozoic and Archean granites (Artemieva et al., 2017), and ratios of Th/U and K/U are mostly in the range typical of Middle and Late Proterozoic granites (Table 1). Most samples show a linear relationship between Th/U and K/U (Fig. 3), indicating that the samples as a group show coherent geochemical behavior and no evidence of significant mobilization of their heat-producing elements.

Three samples show some notable variations, however. Sample 10MSA-2.3 is a red-colored biotite leucogranite with very low U and moderately high Th and K, which results in highly elevated ratios of Th/U and K/U; these high ratios are chiefly a result of the low U concentration in this sample. It appears weathered on the surface but has a zircon $\delta^{18}O$ = 8.2‰ (Goodge et al., 2017), indicating a crustal melt origin with no hydrothermal alteration in the source area. Sample 10MSA-3.5 is a foliated Ms-Bt leucogranite with undetectable U and low Th and K, resulting in an abnormally low concentration of heat-producing elements. This sample has a zircon $\delta^{18}O$ = 5.1‰, indicating a mantle melt origin. Sample 10LWA-6.3 has undetectable U, high Th and average K, resulting in an anomalously high value of heat production as a result of very high Th. This sample is a layered biotite granite with zircon $\delta^{18}O$ = 7.1‰, indicating a crustal melt origin with no hydrothermal alteration in the source area.

## 4.3  Heat production

The geochemical compositions of igneous rocks can be used to determine crustal heat production based on their concentrations of radioactive elements. Heat production ($H_o$) was calculated for these clast samples based on rock density and concentrations of the heat-producing elements U, Th and K by applying two different algorithms:

$$H_o = 10^{-2} * \rho * (9.67CU + 2.63CTh + 3.48CK) \tag{1}$$

$$H_o = \rho * 0.9928CU*H(^{238}U) + 0.0071CU*H(^{235}U) + CTh*H(Th) + 1.19x10^{-4}CK^{40}*H(^{40}K) \tag{2}$$

where $H_o$ is surface heat production ($\mu W\ m^{-3}$), $\rho$ is density (kg $m^{-3}$), CU is the concentration of U (ppm), CTh is the concentration of Th (ppm), CK is the concentration of $K_2O$ (wt%), $H(^{238}U)$ is the heat production from the isotope $^{238}U$ ($9.37x10^{-5}$ W $kg^{-1}$), $H(^{235}U)$ is the heat production from the isotope $^{235}U$ ($5.69x10^{-4}$ W $kg^{-1}$), $H(Th)$ is the heat production from the isotope $^{232}Th$ ($2.69x10^{-5}$ W $kg^{-1}$), and $H(^{40}K)$ is the heat production from the isotope $^{40}K$ ($2.79x10^{-5}$ W $kg^{-1}$). Method 1 was calculated as in Equation (1) from the formula of Rybach (1976, 1988), using values from Hasterok and Chapman (2011). Method 2 uses the formulation of Turcotte and Schubert (2014) as given in Equation (2). Both methods are included here for the purposes of comparison, and in order that the values can be compared with results from other areas that use either of the calculations. Density ($\rho$) was assumed to be 2.7 x $10^3$ kg $m^{-3}$ in all cases. Using Method 1, the igneous clast compositions yield estimates of heat production ranging from 0.25-7.49 $\mu W\ m^{-3}$, with an average of about 2.6 $\mu W\ m^{-3}$ and

1σ standard deviation of 1.9 µW m$^{-3}$ (Table 1). Most of the variation observed in these samples comes from variations in concentrations of U and Th. Method 2 gives quite similar results. It is notable that the two samples with both the highest and lowest calculated heat production (10MSA-3.5 and 10LWA-6.3) have anomalous concentrations of U and/or Th, suggesting that these may represent outliers that are not representative of crust in the glacial catchment area.

Estimates of heat production versus age are plotted in Fig. 4. Compared to an average value for surface heat production in stable continental shield regions of ~2 µW m$^{-3}$ (Jaupart et al., 2016), most of the Antarctic clast samples are of similar magnitude, with 11 of 18 falling between 1-4 µW m$^{-3}$. Some of the values are higher than those reported for other cratonic areas (e.g., Canadian Shield and Grenville Orogen; Mareschal and Jaupart, 2013; Jaupart et al., 2016) and most are higher than the bulk upper crustal average of about 1.6 µW m$^{-3}$ (Kemp and Hawkesworth, 2003; Jaupart et al., 2016). As a group,

the granite clast values overlap a range of 1-3 µW m$^{-3}$ observed in granites globally (Artemieva et al., 2017), and their mean of about 2.6 µW m$^{-3}$ is quite similar to the global average granitic heat production of 2.5 µW m$^{-3}$ (Rybach, 1976; Haenel et al., 1988). The glacial granite clasts overlap significantly with Proterozoic granites worldwide (Artemieva et al., 2017), with average heat production of 3.83 ± 2.14 µW m$^{-3}$ (Fig. 4). Heat production from the clasts is comparable to estimates obtained from Archean and Paleoproterozoic bedrock exposed in the coastal region of southern Prydz Bay (2.4-2.6 µW m$^{-3}$; Carson

and Pittard, 2012; Carson et al., 2013). Four of the clasts give high values between 4.0-7.5 µW m$^{-3}$, which are similar to global occurrences of crust characterized by high heat production (Mareschal and Jaupart, 2013; Jaupart et al., 2016) and exemplified by the Central Australian Heat Flow Province (CAHFP; Neumann et al., 2000; Sandiford and McLaren, 2002; McLaren et al., 2003). Nonetheless, all but two of the samples in this suite have heat production less than the mean for the CAHFP (4.6 µW m$^{-3}$).

The variability in heat production shown by the data presented here resembles that observed in regions comprised by Precambrian shields or granitic batholiths and likely represents real heterogeneities in the source region. Although the precise distribution of heat-producing rocks in the source area from which these clast samples were eroded is not known, this group may collectively provide a qualitatively random sample that provides a means to assess average heat production for a broad region of the continental interior. Compared to examples globally (Mareschal and Jaupart, 2013; Jaupart et al., 2016;

Artemieva et al., 2017), the Proterozoic igneous rocks in this study indicate that heat production in central East Antarctica is like that of typical continental shield areas and demonstrably different from the anomalously warm region represented by the CAHFP. Geological and geophysical correlations between cratonic rocks in southern Australia (Gawler craton) and the Wilkes Land region of East Antarctica (e.g., Oliver and Fanning, 1997; Aitken et al., 2014; Goodge and Finn, 2010; Boger, 2011; Goodge and Fanning, 2010, 2016), have been used as the basis for extrapolating high heat flow values reported for the

CAHFP into East Antarctica (Carson et al., 2013). To date, no direct constraint on terrestrial heat flow has been provided for this area of Wilkes Land, and how far south toward Dome C and the upper Aurora and Wilkes subglacial basins such a province may extend is not clear. However, the data reported here indicate that areas of west-central East Antarctica at least as far north as 80°S may best be characterized as having only modest heat flow.

### 4.4 Heat flow

Geothermal heat flow can be estimated from the empirical relation with crustal heat production (Lachenbruch, 1968; Roy et al., 1968). In the absence of direct terrestrial heat flow measurements, as is the case for Antarctica, it is possible to calculate heat flow from heat production by assuming a thickness of the upper crustal heat-producing layer (Sandiford and McLaren, 2002; Turcotte and Schubert, 2014). This thickness, $h_r$, is the length scale for decrease in $H_o$ with depth in the upper crust (where most heat-producing elements are concentrated) and is determined from the slope of the function linking heat flow and heat production (q-H). Although $H_o$ is thought to decrease exponentially with depth (Lachenbruch, 1968), a first-order estimate of terrestrial heat flow can be obtained from:

$$q_o = q_m + q_r + (H_o * h_r) \tag{3}$$

where $q_o$ is the surface heat flow (mW m$^{-2}$), $q_m$ is the mantle heat flow, $q_r$ is the 'reduced' heat flow contributed by heat production in the middle and lower crust, and other terms are as defined above. For stable Precambrian continental crust, average values for $q_m$ are about 14 mW m$^{-2}$ and $q_r$ is about 15 mW m$^{-2}$ (Sandiford and McLaren, 2002; Perry et al., 2006; Levy et al., 2010; Mareschal and Jaupart, 2013; Jaupert et al., 2016). Based on similarities in age and thickness to the Canadian and Scandinavian shields, a value of 7.3 km for $h_r$ is used here. Using the relationship above and heat production results, the surface terrestrial heat flow is estimated from the igneous clast population to range from about 31-84 mW m$^{-2}$ (Table 1), with an average of 48.0 ± 13.6 mW m$^{-2}$ (1σ standard deviation; Fig. 5). The average value may be regarded as an integrated estimate of heat flow across the area of erosion within the catchment, but it is probably a maximum because it is derived from values of heat production that are biased to crustal granites.

### 4.5 Uncertainties

Because estimates of heat flow are used in ice-sheet models, it is important to consider uncertainties in the values used as input parameters. Here I consider uncertainties in the estimates of heat production and heat flow provided above.

#### 4.5.1 Uncertainties in $H_o$

Laboratory precision on elemental analyses is very high (instrumental precision within 0.2% for $K_2O$ by XRF and within 2% for U and Th by ICPMS), density is assumed, and constants of heat-production for various elements are assumed. Therefore, individual uncertainties for $H_o$ were not calculated because they are expected to be very low relative to other parameters involved in calculation of heat flow.

#### 4.5.2 Uncertainties in $q_o$

Uncertainties in the linear relationship used to calculate surface heat flow ($q_o$) can be modeled using the following expression:

$$\Delta q_o = \Delta q_m + \Delta q_r + (\overline{H}_o * \Delta h_r) + (\overline{h}_r * \Delta H_o) \tag{4}$$

where $\Delta q_o$ is the sum of uncertainties represented by the variables included in equation (3). Determining reasonable values for most of the $\Delta$ terms is problematical because the corresponding terms in the heat-flow equation are either based on model-derived values or are simply poorly constrained by limited empirical data. Because geological and seismological data indicate that East Antarctica is a stable craton, we can use typical cratonic values for $q_m$ and $q_r$ as a basis for evaluating uncertainty in these terms. For this analysis, $\Delta q_m$ is taken to be $\pm 2.5$ mW m$^{-2}$ based on a compilation of estimates worldwide for stable continental shield areas that range mostly from 12-17 mW m$^{-2}$ (Mareschal and Jaupart, 2013; Jaupart et al., 2016). Uncertainty in the lower-crustal term, $\Delta q_r$, is taken to be 3.0 mW m$^{-2}$, assumed as a general variance ($\pm 20\%$) around a representative value of 15 mW m$^{-2}$ for lower-crustal heat flow. A mean value of $\overline{H}_o = 2.6$ µW m$^{-3}$ is used from the data reported here and the representative average value of $\overline{h}_r = 7.3$ km that was used to calculate heat flow is assumed here. Uncertainty in heat production, $\Delta H_o$, is taken as a $1\sigma$ standard deviation of the calculated values (1.86 mW m$^{-2}$), and uncertainty in the length scale, $\Delta h_r$, is assumed to be 1500 m ($\pm 20\%$), corresponding to the magnitude of subglacial topographic relief along the transport direction within the glacial source area catchment. Based on these inputs, we can derive a general uncertainty for the surface heat flow term ($\Delta q_o$) of about 23 mW m$^{-2}$ (Fig. 5). This is a large value compared to the nominal mean value of 48 mW m$^{-2}$ obtained here, and it reflects large natural variability in lithosphere properties as well as few direct constraints on mantle heat flow, lower crustal heat flow, and the vertical distribution of heat-producing elements in continental crust. Of this estimated uncertainty, 24% is contributed by the $\Delta q_m$ and $\Delta q_r$ terms, and 76% is attributed to the multiplying effects of the thickness and uncertainty of the upper-crustal heat-producing layer ($\overline{h}_r$ and $\Delta h_r$). Only 8% is contributed by $\Delta H_o$ itself. Together, the large combined uncertainty is therefore contributed mainly by mantle heat flow, lower crustal heat flow, and the vertical distribution of heat-producing elements; conversely, estimates of upper crustal heat production from the glacial clast samples are not an important source of uncertainty. Nonetheless, the overall range in surface heat flow covered by this uncertainty is consistent with the range of values reported for other cratons, lending support to the idea that the recovered glacial clasts are indeed representative of heat flow known from typical Archean and Proterozoic shield areas. Despite the inherent large uncertainties, the first-order results can help to inform future ice-sheet modeling.

## 5 Discussion

The glacial igneous clasts sampled for this study indicate that upper crustal heat production for at least a part of central East Antarctica is in the range of 0.3-7.5 µW m$^{-3}$, with average value of $2.6 \pm 1.9$ µW m$^{-3}$ (n = 18). Assuming typical values of mantle heat flux, lower-crustal heat flux, and an upper-crustal length factor appropriate for stable continental cratons, the derived heat production corresponds to an average surface heat flux of 48 mW m$^{-2}$. This approach assumes typical cratonic values for mantle and lower-crustal contributions, which it is reasonable given what is known about East Antarctic lithosphere (e.g., An et al., 2015). The net upper crustal contribution to surface heat flow is therefore about 19 mW m$^{-2}$.

Although clasts eroded from the subglacial bedrock surface represent a close approach to a random sampling of continental crust in East Antarctica, it is certainly possible that other rocks buried more deeply beneath the glacial interface in the upper or middle crust may harbor high heat-producing elements. In such a case, the distribution of heat-producing elements with depth may yield a greater total crustal contribution to heat flow. Lacking specific constraints to the contrary, however, a conservative approach is to assume a distribution of heat-producing elements based on analysis and models from other similar cratons. Several lines of evidence indicate that upper continental crust in most cratons is dominated by granites (study of exposed basement, borehole data, seismology; Artemieva et al., 2017), which are unique in having high concentrations of heat-producing elements U, Th and K (Jaupart and Mareschal, 2003). This can yield an order of magnitude greater heat production compared to granulites, gabbros, and amphibolites of the middle and lower crust (Artemieva et al., 2017). In general, granites in the upper crust therefore provide the greatest contribution to surface heat flow. If the Mesoproteozoic and Paleoproterozoic granitic samples of this study are representative of upper continental crust in cratonic East Antarctica, they likely provide a significant crustal contribution to surface heat flow.

Despite a small sample size, the results here are considered to be representative of crust in central East Antarctica. First, it is important to note that the collection process was as randomized as possible given the time limitations at each site. All igneous clasts with potential age and geochemical signature were sampled, providing a large composite sample set. Second, the samples screened for detailed petrologic analysis have a wide age distribution, are well characterized in terms of geochemistry and isotopic composition, and comprise distinct petrogenetic groups (see Goodge et al., 2017). That is, they are not cogenetic or derivative from one another but rather representative of heterogeneous crust. Third, none of the clast ages are known from other areas of bedrock exposure in the Transantarctic Mountains or along the greater Wilkes Land margin, such that they appear to represent a heretofore unrecognized and unique cratonic igneous terrain. At a minimum, the results obtained from this sample suite apply to the source area indicated on Figure 2. Extrapolation over a broader area is unconstrained but may include some or all of the greater Byrd Glacier drainage network, perhaps extending as far north as Dome C. Although the data provided in this study are thought to be representative of crust in the interior of west-central East Antarctica, it is not possible to resolve gradients in geothermal properties within the sampled drainage area. For example, a comparison of samples at sites MSA (n = 3) and LWA (n = 10) shows no discernable pattern in age, heat production, or heat flow. Lacking a higher sampling density, the small clast sample size, sample age variation, and heterogeneity of bedrock geology underlying the Transantarctic Mountains make it difficult to distinguish gradients in either heat production or heat flow across individual drainages.

The total surface heat flux is quite similar to the average heat flux of 53 mW m$^{-2}$ from 13 cratonic shield provinces globally (Jaupart et al., 2016). Likewise, Nyblade and Pollack (1993) found average surface heat flow values of 42 mW m$^{-2}$ for Archean provinces and 47 mW m$^{-2}$ for Paleoproterozoic provinces, which represents a general depletion of heat-producing elements in continental crust with increasing age. The heat flow results obtained here are also similar to earlier estimates for East Antarctica determined by geophysical modeling and inversion of ice borehole temperature profiles, which indicate a broad region with low to moderate values of 50-60 mW m$^{-2}$ (Shapiro and Ritzwoller, 2004; Fox Maule et al.,

2005). An et al. (2015) used a 3-D S-wave velocity model to construct temperature profiles for Antarctic lithosphere, from which they derived an average surface heat flux of 47 mW m$^{-2}$ for the Gamburtsev province. This is lower than the average of 57 mW m$^{-2}$ proposed by Shapiro and Ritzwoller (2004) or East Antarctica, but quite comparable to the estimate provided here.

Taken together, the heat production and surface heat flow values estimated for the glacial igneous clasts discussed here appear to be representative of typical Archean-Proterozoic cratonic lithosphere. As a group they are distinctly different from the regional pattern shown by anomalously warm Proterozoic crust in central Australia with average $q_o$ = 80 mW m$^{-2}$ (McLaren et al., 2003), which has been suggested to extend across the Wilkes Land margin of Antarctica based on Gondwana supercontinent reconstructions (Carson et al., 2014; Aitken et al., 2014). Despite general age similarities among

some of the clast population with parts of the Gawler Craton, and basement age correlations that indicate continuity of Mawson-type crust into the Wilkes sector of East Antarctica (Goodge and Fanning, 2016), the proxy heat production determinations and heat flow estimates provided here suggest that central portions of the East Antarctic ice sheet are underlain by stable continental crust with quite normal thermal properties represented by average values of heat production of about 2.5 μW m$^{-3}$ and heat flow of about 50 mW m$^{-2}$.

Estimates of terrestrial heat flow such as those provided here can also be used to assess the effect of heat flow on ice-sheet mass balance. For example, Pollard et al. (2005) evaluated the effect of varying heat flow regimes on ice-sheet behavior by modeling changes in Antarctic ice volume, ice-sheet surface elevation, and area of the base at its pressure-melting point as a function of differing heat-flow regimes. Their models used three different geothermal heat flow distributions: (a) uniform heat flow of 37.7 mW m$^{-2}$, representing typical values of Archean cratons; (b) uniform at 75.4 mW

m$^{-2}$, to mimic Proterozoic lithosphere characterized by high crustal heat production; and (c) spatially varying heat flow based on the distributions of different crustal provinces extrapolated from craton-margin geology, and including values of 41 and 55 mW m$^{-2}$ across most of East Antarctica. The values of heat production and heat flow estimated for central East Antarctica in this study are most consistent with their third approach; the average heat flow value of the Proterozoic granitoid samples is higher than in the case of uniform Archean lithosphere, yet lower than that assumed for Proterozoic lithosphere with high

crustal heat production. Because the modeling of Pollard et al. (2005) shows a large effect of heat flow on the area of the ice-sheet base at its pressure-melting point, inputting appropriate values of crustal heat flow is vitally important for predicting, for example, the thermal and physical conditions of the basal ice-sheet regime.

To provide a simple model for the distribution of heat flow across the catchment area sampled in this study, mean heat flow values were calculated in two ways (Table 2). First, the set of 18 samples was divided into equal quintiles representing

ranges of 10 mW m$^{-2}$ each. Average heat flow values were calculated for each quintile, as was a percentage of the measurements falling in that range (Fig. 6a). Each quintile thus represents a proportionally-based average heat flow value that could be used as an input for ice sheet models. Assuming that the igneous and metamorphic crust beneath the East Antarctic ice sheet is heterogeneous in age and composition, this proportional distribution of heat flow values may better reflect the complexities of crustal geothermal input as a function of subglacial area compared to a simple average. Second,

the samples were grouped by age and average heat flow values calculated for each of four groups (Fig. 6b). This approach provides a reasonable estimate of heat flow potentially contributed by igneous crust proportionally represented by different age groups. Although the sample values were divided arbitrarily into five groups using the first method, this approach shows that about 61% of the sample results are <50 mW m$^{-2}$ (also indicated by the skewed distribution of values in Figure 5), indicating that the bulk of crust underlying the East Antarctic ice sheet has relatively low long-range average heat flow. The second approach, perhaps more useful from a modeling perspective because it groups samples by age, illustrates that for individual age groups the values are also quite modest, ranging from about 42-55 mW m$^{-2}$ and similar to the total group average. It is noteworthy that this range is nearly identical to the heterogeneous heat-flow model adopted by Pollard et al. (2005), appearing to validate the earlier study. Future ice-sheet stability modeling combined with the estimates of low to intermediate sub-glacial heat flow found in this study may thus help to further refine predictions of ice-sheet behavior.

## 6 Conclusions

Based on geochemical analysis of a suite of glacially-eroded granitic rock clasts, average heat production from an inferred large Proterozoic igneous crustal province in central East Antarctica is estimated to be about 2.5 $\mu$W m$^{-3}$, and the corresponding average surface heat flow is about 48 mW m$^{-2}$. These geothermal properties are quite similar to average Archean and Proterozoic cratonic shields globally, despite being biased here to granitic compositions. Although the source of the granite clasts is not precisely known, they were likely derived from a region extending into central East Antarctica from near the inlet to Byrd Glacier. This region contrasts with other areas marked by high heat flow, such as the Central Australia Heat Flow Province and some parts of East Antarctica near Prydz Bay, indicating that crust in those areas likely does not extend into central regions of the continental interior.

Heat flow as estimated in this study is valuable for several reasons. First, the values obtained here are similar to an estimate of heat flow derived by modeling of a borehole temperature profile near Dome C (54 mW m$^{-2}$; Fischer et al., 2013), helping to validate the earlier model finding. Likewise, they are consistent with the general range of values indicated by inversion of geophysical data from cratonic East Antarctica (e.g., Shapiro and Ritzwoller, 2004; Fox Maule et al., 2005; An et al., 2015). The average value of heat flow determined in this study (48 mW m$^{-2}$) is quite similar to that obtained by An et al. (2015) from inversion of recent high-quality S-wave data in central East Antarctica (47 mW m$^{-2}$). In detail, the values obtained here show a similar range to those indicated in the model derived from magnetic data (Fox Maule et al., 2005), both of which indicate that lithologic and, therefore, geothermal variations are real. Second, the new data provide a unique estimate of heat production and terrestrial heat flow that can be used as an input to ice-sheet stability models. In particular, they validate the general approach by Pollard et al. (2005) in which basal heat flow is varied by area depending on age and character of the subglacial geology. There is similar variability within this sample group that probably reflects the lithologic heterogeneity to be expected in continental shields. Third, although the data presented here provide a good approximation of both heat production and heat flow in an otherwise inaccessible region of East Antarctica, the existing uncertainties

associated with extrapolating heat flow from heat production illustrate the critical need for precise *in-situ* measurement of terrestrial heat flow from the subglacial environment. One attempt to do so beneath the Whillans Ice Stream in West Antarctica (Fisher et al., 2015) measured a heat flux of 285 mW m$^{-2}$. This extraordinarily high value, even greater than that observed on modern ocean ridges (typically ~100-250 mW m$^{-2}$ near the ridge axis and one third of that for oceanic crust >50

5  Ma; Stein, 1995), likely is perturbed by advective heat transfer associated with subglacial flow of water and is therefore not representative of terrestrial heat flow in West Antarctica. A more recent measurement of 88 mW m$^{-2}$ obtained in subglacial sediment near the grounding zone of the Whillans Ice Stream provides a better constraint on geothermal heat flow in West Antarctica that contrasts with estimates for cratonic East Antarctica (Begeman et al., 2017). Despite the difficulty in obtaining reliable heat flow data from the subglacial environment, it should be a high research priority that can be addressed

by drilling through the ice sheets at as many sites as possible in order to assess crustal heterogeneity. Last, these estimates of low to moderate crustal heat flow indicate that some large regions of the interior East Antarctic ice sheet may be expected to be frozen at the bed, which is of use to future drilling projects that plan to intersect the glacial bed.

**Data availability.** Data supporting the conclusions are listed in Table 1.

**Sample availability.** Samples referred to in this study are housed at the University of Minnesota Duluth and available on
request to the author.

**Competing interests.** The author declares that he has no conflict of interest.

**Acknowledgments.** Field and analytical portions of this project were supported by the National Science Foundation (award 0944645). Jacqueline Halpin and Jean-Claude Mareschal provided helpful feedback on the approach to estimating heat production and heat flow, and Jeff Severinghaus kindly reviewed an earlier draft manuscript. John Swenson generously
provided insight into the treatment of uncertainties.

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

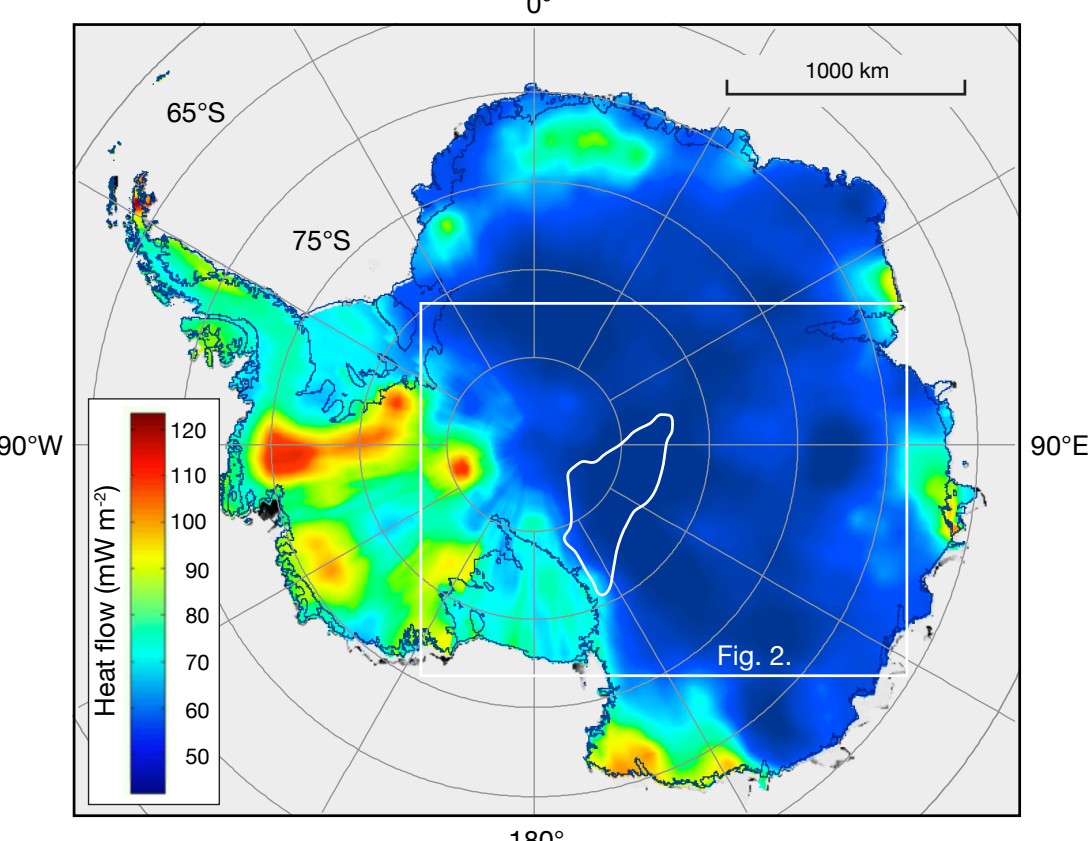

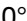

**Figure 1: Terrestrial heat flow in Antarctica, from the mean geothermal heat flux model of Van Liefferinge and Pattyn (2013), which averages heat flow determined from multiple geophysical datasets. Inset white box shows area of Figure 2, including glacial drainage sourcing bedrock igneous rock clasts (white outline).**

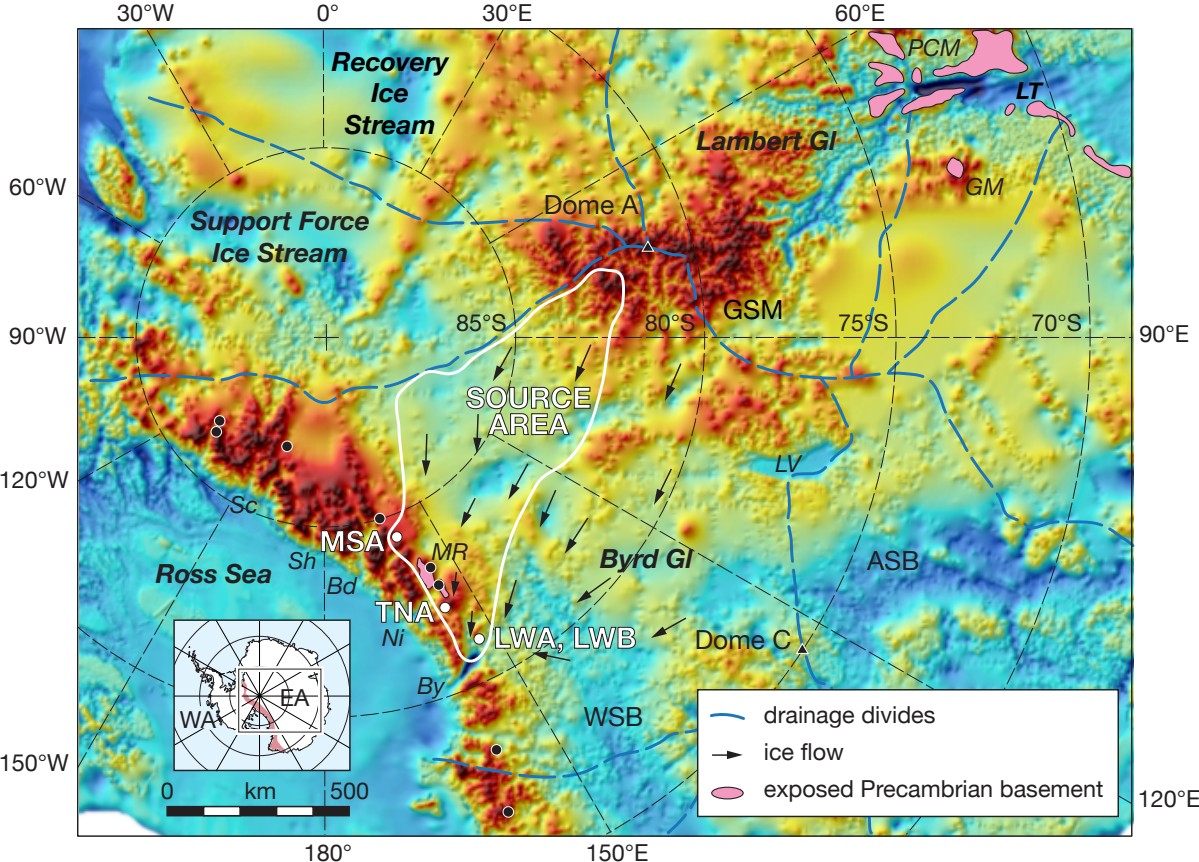

**Figure 2: Map showing potential source areas for dated glacial igneous clasts, superimposed on the Bedmap2 subglacial topography of East Antarctica (Fretwell et al., 2013). Principal features are ice-sheet catchment areas (marked by thin blue drainage divides), ice flow directions in the broad Byrd Glacier drainage (arrows; Rignot et al., 2011), and areas of Precambrian basement exposure (pink). Composite source area (outlined by heavy white line) was determined from the ice flow-fields that contribute ice to each of the sample sites (white circles). Other sampled sites shown by black circles. Because transport distance is not known for any of the individual clasts, possible bedrock sources could lie anywhere between the sample sites and the top of the ice-shed overlapping the Gamburtsev Subglacial Mountains (GSM). Sample sites: LWA, Lonewolf Nunataks; MSA, Mt. Sirius; TNA, Turret Nunatak. Other abbreviations:  ASB, Aurora Subglacial Basin; GM, Grove Mountains; LT, Lambert trough; LV, Lake Vostok; MR, Miller Range; PCM, Prince Charles Mountains; WSB, Wilkes Subglacial Basin. Outlet glaciers:  Bd, Beardmore; By, Byrd; Ni, Nimrod; Sc, Scott; Sh, Shackleton.**

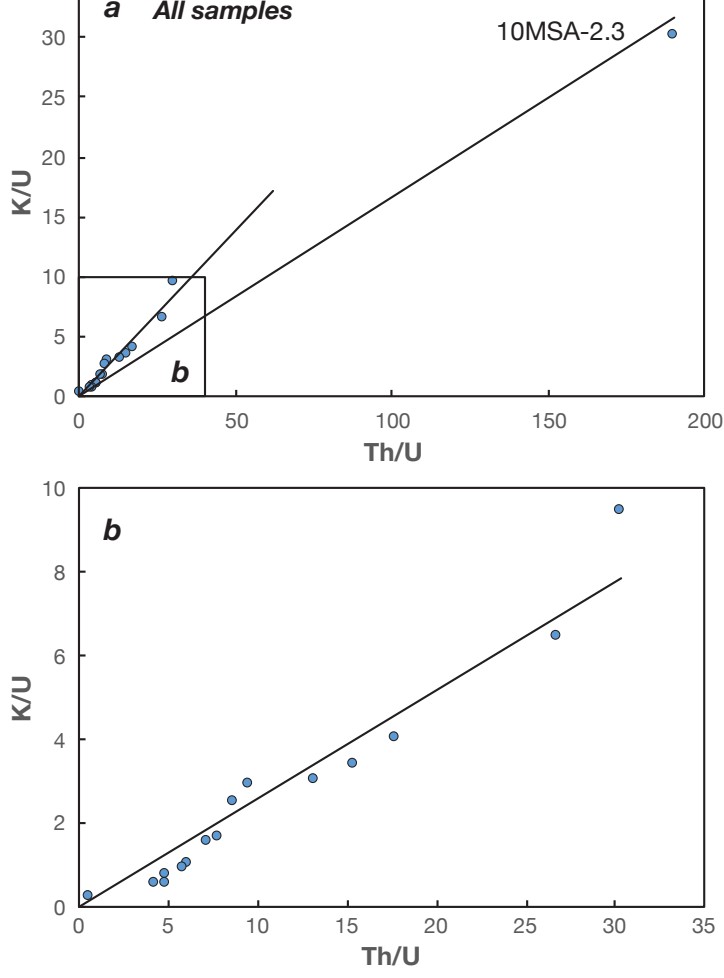

**Figure 3. (a) Plot of Th/U vs. K/U in glacial igneous clasts, with detail in (b) that excludes sample 10MSA-2.3. Linear regression in (b) was calculated for all samples minus sample 10MSA-2.3.**

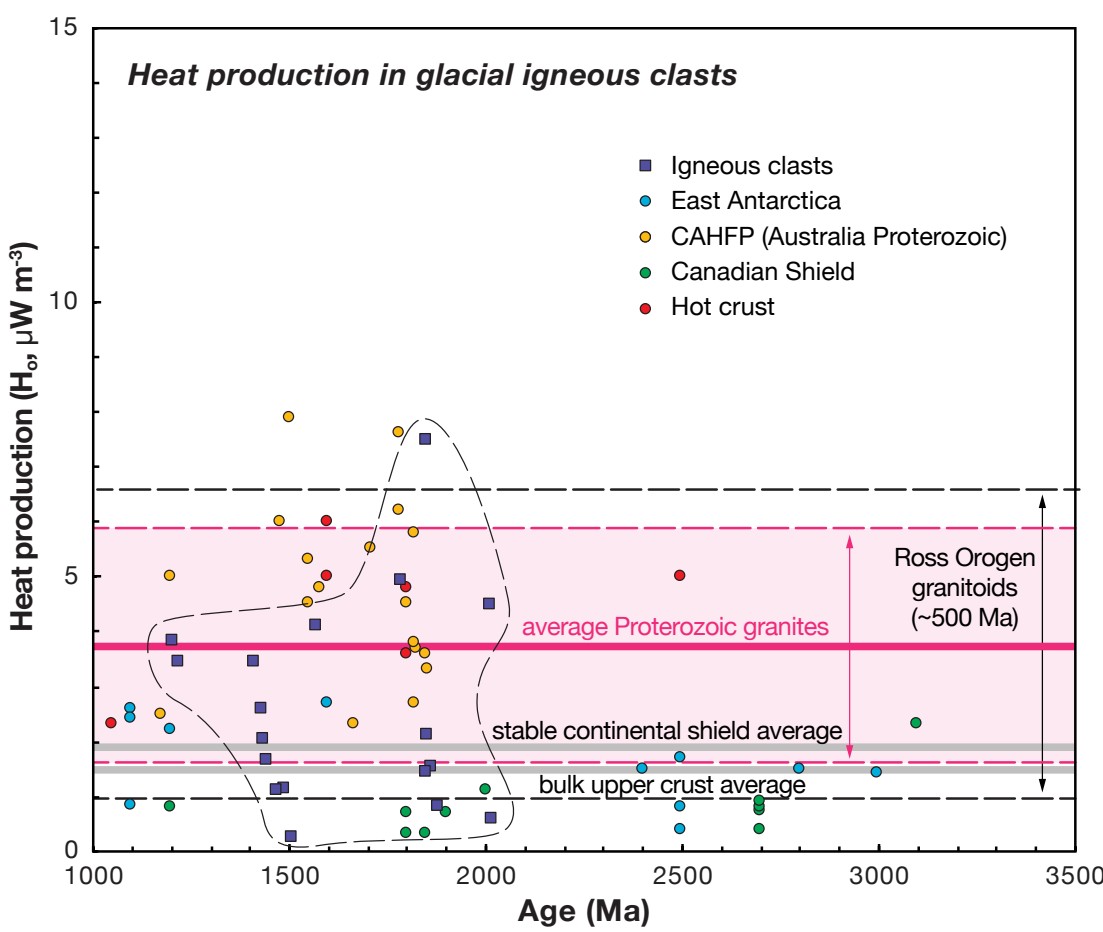

**Figure 4.** Plot of surface heat production ($H_o$) vs. age for igneous glacial clasts. Data listed in Table 1. Range of heat production values in Ross Orogen granites from unpublished data. For comparison, values are shown from East Antarctica (Carson and Pittard, 2012; Carson et al., 2014), the Central Australian Heat Flow Province (CAHFP; McLaren et al., 2003), the Canadian Shield (Mareschal and Jaupart, 2013; Jaupart et al., 2016), and areas of high heat production in stable continental provinces ('hot crust'; Mareschal and Jaupart, 2013; Jaupart et al., 2016). Average heat production in Middle and Late Proterozoic granites of about 3.8 µW m$^{-3}$ from Artemieva et al. (2017). Bulk upper crustal average of about 1.6 µW m$^{-3}$ from Kemp and Hawkesworth (2003).

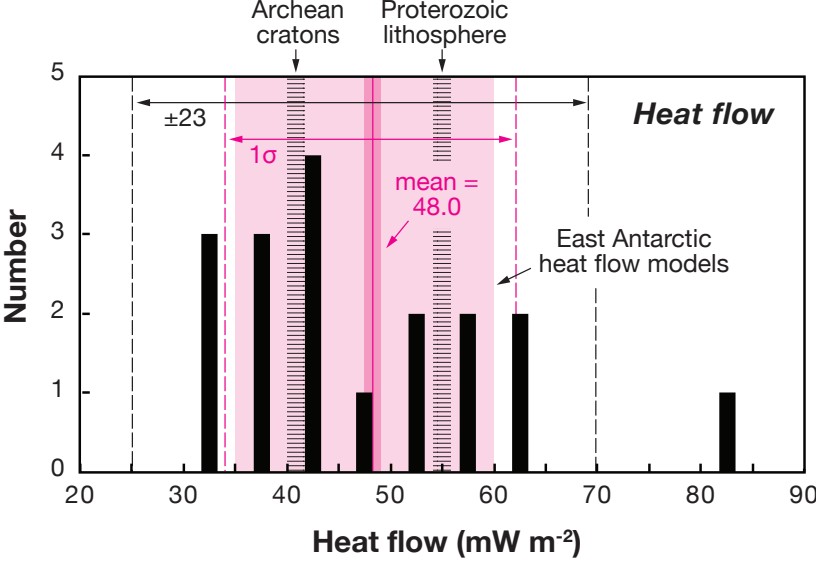

**Figure 5. Histogram of heat flow values estimated from heat production in the glacial clasts. Mean value (magenta) is 48.0 mW m$^{-2}$ (n = 18) with a 1σ standard deviation of 13.6 mW m$^{-2}$. Consideration of uncertainties in calculation of heat flow indicates an overall uncertainty of about ±21 mW m$^{-2}$ (see text). Range of heat flow modeled for East Antarctica shown for comparison (light pink; Van Liefferinge and Pattyn, 2013). Global average values for Archean cratons and Proterozoic lithosphere shown by ruled bars (Nyblade et al., 1999).**

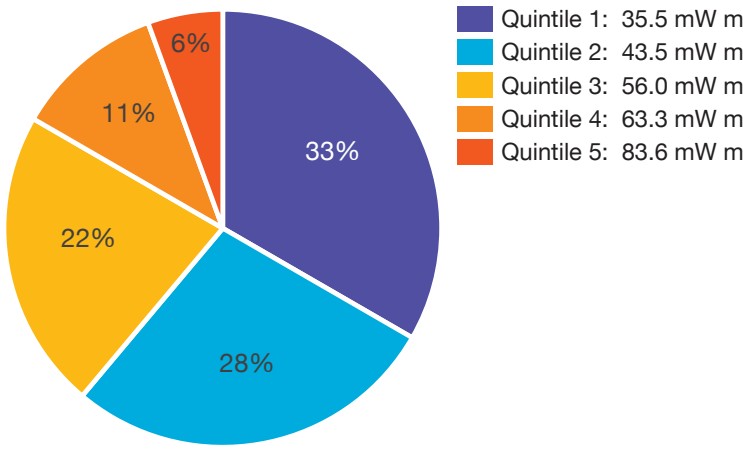

*a* Heat flow distribution by quintile

| | |
|---|---|
| Quintile 1: | 35.5 mW m⁻² |
| Quintile 2: | 43.5 mW m⁻² |
| Quintile 3: | 56.0 mW m⁻² |
| Quintile 4: | 63.3 mW m⁻² |
| Quintile 5: | 83.6 mW m⁻² |

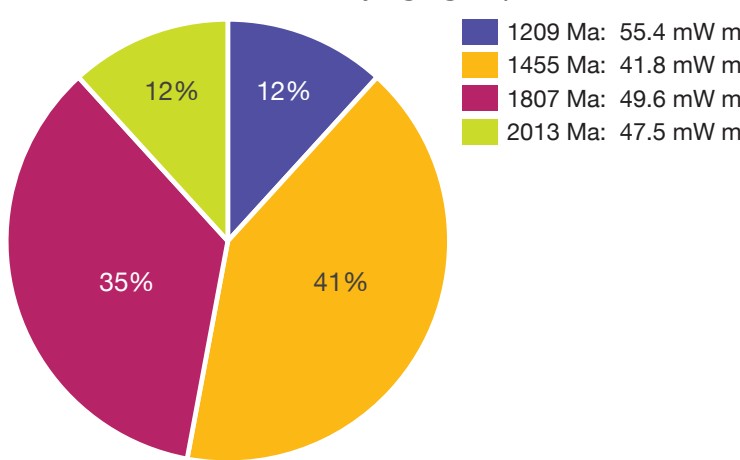

*b* Heat flow distribution by age group

| | |
|---|---|
| 1209 Ma: | 55.4 mW m⁻² |
| 1455 Ma: | 41.8 mW m⁻² |
| 1807 Ma: | 49.6 mW m⁻² |
| 2013 Ma: | 47.5 mW m⁻² |

**Figure 6: Summary pie diagrams showing distribution of heat flow estimates from this study. (a) Distribution of heat flow by quintiles between 30-80 mW m⁻². Quintile averages shown, with the highest quintile represented by one sample with calculated heat flow of about 84 mW m⁻². (b) Distribution of average heat flow by age groups. Values shown in Table 2.**

**Table 1. Estimates of $H_o$ and $q_o$ for igneous clast samples, central East Antarctica.**

| Sample [a] | Age (Ma) | U (ppm) | Th (ppm) | Th/U | $K_2O$ (wt%) | K (wt%) | K/U | Density (g cm$^{-1}$) | $H_o$ (Method 1) [b] ($\mu$W m$^{-3}$) | $H_o$ (Method 2) [b] ($\mu$W m$^{-3}$) | $q_o$ [c] (mW m$^{-2}$) |
|---|---|---|---|---|---|---|---|---|---|---|---|
| 10LWA-13.1 | 1204 | 5.7 | 27.7 | 4.9 | 3.80 | 3.15 | 0.55 | 2.7 | 3.81 | 3.79 | 56.8 |
| 10LWA-11.1 | 1213 | 2.5 | 32.9 | 13.2 | 4.72 | 3.92 | 1.57 | 2.7 | 3.43 | 3.40 | 54.1 |
| 10MSA-2.3 | 1410 | 0.2 | 38.0 | 190.0 | 7.26 | 6.03 | 30.13 | 2.7 | 3.43 | 3.35 | 54.1 |
| 10TNA-1.1 | 1430 | 1.2 | 18.5 | 15.4 | 4.39 | 3.64 | 3.03 | 2.7 | 2.04 | 1.98 | 43.9 |
| 10LWA-6.5 | 1432 | 2.7 | 15.8 | 5.9 | 8.16 | 6.77 | 2.51 | 2.7 | 2.59 | 2.46 | 47.9 |
| 10LWB-4.3 | 1448 | 1.2 | 11.4 | 9.5 | 5.76 | 4.78 | 3.99 | 2.7 | 1.66 | 1.57 | 41.2 |
| 10LWB-3.8 | 1470 | 1.2 | 5.8 | 4.8 | 4.19 | 3.48 | 2.90 | 2.7 | 1.12 | 1.05 | 37.2 |
| 10LWA-20.1 | 1486 | 1.2 | 8.6 | 7.2 | 2.35 | 1.95 | 1.63 | 2.7 | 1.14 | 1.11 | 37.4 |
| 10MSA-3.5 | 1508 | 0.0 | 1.5 | – | 1.57 | 1.30 | – | 2.7 | 0.25 | 0.23 | 30.9 |
| 10LWA-6.4 | 1570 | 6.4 | 26.9 | 4.2 | 5.65 | 4.69 | 0.73 | 2.7 | 4.11 | 4.05 | 59.0 |
| 10LWA-14.1 | 1786 | 5.4 | 41.9 | 7.8 | 5.77 | 4.79 | 0.89 | 2.7 | 4.93 | 4.89 | 65.0 |
| 10LWB-4.5 | 1848 | 0.5 | 13.4 | 26.8 | 3.89 | 3.23 | 6.46 | 2.7 | 1.45 | 1.39 | 39.6 |
| 10LWA-6.3 | 1850 | 0.0 | 98.3 | – | 5.38 | 4.46 | – | 2.7 | 7.49 | 7.54 | 83.6 |
| 10LWA-7.1 | 1854 | 0.6 | 18.2 | 30.3 | 6.83 | 5.67 | 9.44 | 2.7 | 2.09 | 1.99 | 44.3 |
| 10LWB-4.1 | 1865 | 1.2 | 10.4 | 8.7 | 4.92 | 4.08 | 3.40 | 2.7 | 1.51 | 1.44 | 40.1 |
| 10MSA-3.3 | 1876 | 1.9 | 1.0 | 0.5 | 2.29 | 1.90 | 1.00 | 2.7 | 0.78 | 0.74 | 34.7 |
| 10LWA-10.1 | 2010 | 2.9 | 51.3 | 17.7 | 0.73 | 0.60 | 0.21 | 2.7 | 4.47 | 4.54 | 61.6 |
| 10LWA-8.1 | 2015 | 0.8 | 4.9 | 6.1 | 0.53 | 0.44 | 0.55 | 2.7 | 0.61 | 0.61 | 33.4 |
| Mean | | 2.0 | 23.7 | 22.1 | 4.34 | 3.60 | 4.31 | | 2.61 | 2.56 | 48.0 |
| Std dev | | 2.0 | 23.6 | – | 2.18 | 1.81 | – | | 1.86 | 1.89 | 13.6 |
| Prot. average [d] | | 2.4 | 10.0 | 3.63 | | 2.39 | 0.98 | | 3.83 | 3.83 | |

[a] Samples collected at the following sites: LWA and LWB, Lonewolf Nunataks (2 sites); MSA, Mt. Sirius; TNA, Turret Nunatak (see Goodge et al., 2017).

[b] Heat production ($H_o$) was calculated from geochemical analysis in two ways. Method 1 uses the relation $H_o = 10^{-2} * \rho * (9.67 [U] + 2.63 [Th] + 3.48 [K])$, using values after Rybach (1988) and Hasterok and Chapman (2011). Method 2 uses the relation $H_o = (0.9928 [U] * H(U^{238}) + (0.0071 [U] * H(U^{235}) + ([Th] * H(Th)) + 1.19e10^{-4} * [K] * H(K^{40})) * D$ (Turcotte and Schubert, 2014), where D is density. Both assume an average density for granitic rocks of 2.7 g cm$^{-3}$.

[c] Surface heat flow ($q_o$) determined from $q_o = q_m + q_r + (H_o \cdot h_r)$ (Turcotte and Schubert, 2014). Moho heat flux ($q_m$) is assumed to be 14 mW m$^{-2}$ for stable continental shield areas (Mareschal and Jaupart, 2013; Jaupart et al., 2016), lower crustal heat flow ($q_r$) is assumed to be 15 mW m$^{-2}$, and length scale for reduction in heat production with depth ($h_r$) is assumed to be 7.3 km.

[d] Artemieva et al. (2017)

**Table 2: Heat flow estimates in proportions based on quintile ranges and age.**

| Binned by quintile | | | | Binned by age | | | |
|---|---|---|---|---|---|---|---|
| | Heat flow | | | | Heat flow | | |
| Quintile | (mW m$^{-2}$) | No. | % | Age (Ma) | (mW m$^{-2}$) | No. | % |
| 1 | 35.5 | 6 | 0.33 | 1209 | 55.4 | 2 | 0.12 |
| 2 | 43.5 | 5 | 0.28 | 1455 | 41.8 | 7 | 0.41 |
| 3 | 56.0 | 4 | 0.22 | 1770 | 49.6 | 6 | 0.35 |
| 4 | 63.3 | 2 | 0.11 | 2013 | 47.5 | 2 | 0.12 |
| 5 | 83.6 | 1 | 0.06 | | | | |