# Peer review of "Crustal heat production and estimate of terrestrial heat flow in central East Antarctica, with implications for thermal input to the East Antarctic ice sheet"

_The Cryosphere, 2017_

## Referee Comment (RC1) · Anonymous Referee #1 · 11 Sep 2017

This article written by J. Goodge is an attempt to improve our knowledge on the geothermal flux under the Antarctic Ice Sheet, and more precisely on the Byrd and Nimrod catchments. This topic is an important issue in our community, regarding the stability of the Antarctic Ice Sheet from a thermal and mechanical point of view in the future. Even if the conclusions do not change the previous estimations of the geothermal flux, the sample-based method used here constitutes a crucial cross-checking from field work. While our knowledge is mainly based on modelling results and a few analysis of temperature profiles in boreholes, this work takes advantage of glacial rock clasts gathered at the catchment output and taken as samples of the whole catchment. This original approach is valuable considering the difficulty to access the bedrock under the

ice sheet. Furthermore, it allows the author to have an interesting interpretation of the heat flow distribution from a geological point of view.

The article is pleasant to read, and its structure makes the text easy to follow. Scientific context, methods, assumptions and discussion are well stated. The figures are precise and complete. The uncertainties are well evaluated and their origin discussed.

General comments

Hereafter are a few remarks that could help improve the manuscript. The main concern I have is related to how the sampling process is explained.

Samples

Your results and interpretation are based on the processing of 18 samples. Of course, the processing step is heavy and it is difficult to have more samples. However, the interpretation of the distribution is based on these 18 samples only, and a different sampling could lead to significantely different interpretation. For example, only one sample correspond to a heat flow higher than 80 mw/m2, and appears like an outlier, but this could appear quite different with a larger sampling. I suggest that a small paragraph discuss this point, even if it is impossible to have more numerous samples.

More serious, I do not understand how the samples have been chosen, and why the different sampling sites are not more equally distributed. This is a problem since your conclusions are very sample-dependent. Furthermore, Table 1 is not so clear, since the acronyms LWA, LWB, MSA, TNA are never explained. I understand it refers to the sites, but I cannot figure out which one for LWB. Why Lonewolf Nunataks would be present so many times, whereas AGA and MRA disappeared ? Did you sampled some rocks at AGA and MRA, and discarded them in a second step ? I think §2 needs to be completed, so that your selection criteria are clearly stated.

Interpretation

Your sampling concerns the Byrd catchment mainly, which, I think, stands across Victoria Land, Adélie Land and Wilkes Land. But your interpretation concerns "the Wilkes Land region" (P.7 L.10), which is a much larger region than your source area. This is a bit confusing. Your conclusion say that an extrapolation of CAHFP under Antarctica is potentially erroneous, whereas your samples concerns a very small part of the Wilkes Land. For example, an extrapolation could be justified close to the coast of the Wilkes Land, or even at Dome C. I suggest the end of §4.3 is changed for a more careful formulation, conclusion and abstract as well.

Exploiting the ice flow ?

I was first very pleased to see that you mixed geology and ice flow aspects. However, after reading the whole paper, I felt like disappointed that you cannot fully exploit the geographical aspect given by the ice flow. The Transatarctic mountains make the problem much complex, but having a comprehensive ice-flow interpretation could be an interesting scientific perspective... to add in conclusion ? In the meantime, could you at least look if there is any gradient between MSA and LWA for example, in heat flow or chemical composition ? Even if it does not give any additional information, you could just mention that you looked at it.

Minor revisions

P2, L12 : Is the work of Fisher et al (2015) not already a direct subglacial measurement of geothermal flux ? So I would say "..., and very few from direct subglacial measurements". The same in the abstract.

P2, L24 : Since the heat budget of basal ice depends both on ice thickness and geothermal flux, there is no direct pure correlation between ice thickness and basal age, I suggest "relatively thick, slow-moving...".

P3, L19 : the samples have travelled for hundred of thousands of years, and the ice sheet has not always had the present shape. In particular during glacial areas, do you have any idea of the shape of the catchment given by continental model outputs ? (if

possible)

P4, L11 : how are the samples selected, and how many for each site ? Why are some sites over-represented in the number of samples ? How is the choice of the sample number affect the conclusion ?

P6, L21 : Could you briefly explain why it is necessary to make use of the two methods, since the results are similar ? I guess that, in other circumstances, they could significantely differ ?

P7, L8 : "Compared to global examples". To make it more clear, could you briefly specify, or add references ?

P7, L19 : I think q and H are not defined, maybe change for a specification in words : "the function linking heat flow and heat production".

P8, L23 : A suggestion concerning $\Delta$hr: could you consider the variability of the bed topography on the source area as an estimator for the crust uncertainty, so that your uncertainty has a better specified origin ? I think you should be close to your present value of 1500 m and will not change your conclusions. If you disagree with this approach, could you better justify this value or add a reference ?

---

## Referee Comment (RC2) · Anonymous Referee #2 · 12 Sep 2017

General comments

Goodge has developed an idea that has appeared in the literature over the past few years regarding the possible impact of high heat producing rocks on sub-glacial heat flow in Antarctica. This is an important problem given both the potentially significant impacts of sub-glacial heat flow variations on ice sheet behaviour, and the established (possibly flawed) approach of using literature averages for surface heat flow in ice sheet models. It is also an interesting area of cross-disciplinary science, and obviously a suitable topic for Cryosphere. Goodge's paper contributes to our understanding of sub-glacial heat flow variation by measuring the heat-producing-element enrichment for a

limited number of assumed bedrock lithotypes sampled from moraine systems. These data are then used to estimate sub-glacial heat flow and a conclusion of average sub-glacial heat flow for East Antarctica is made. The paper is generally well-written and easy to follow but I am a bit concerned that there is insufficient new data, and caveats on the sample size and interpretation method are not adequately outlined. The sample set is very small (five localities and 18 samples) yet the conclusions are extrapolated to draw conclusions about the regional heat flow that are not necessarily justified.

Specific comments

There are many assumptions used in the calculations that rapidly take the reader from simple U, Th and K concentrations of a few eroded rock clasts to calculations of sub-glacial heat flow across a vast area of East Antarctica. Both a mantle heat flow, and a heat contribution from the lower crust are assumed. A length scale for heat production distribution vertically within the crust is also assumed (see further discussion below). Although I fully understand the difficulties of constraining sub-glacial geology, better constraints on mantle heat flow (from seismic velocity data for example) could be used.

The analytical approach for heat-producing element determination is fairly standard and in a geoscience paper would be included only as an appendix. I do understand that for a non-specialist geoscience audience it may be more appropriate in the main text.

U and Th are known to be highly mobile elements in the near surface environment or where there is fluid flow. It would be interesting to look at the Th/U ratios (can this be added to Table 1) and the issue of mobilisation (loss or gain) of these elements to help establish that these values reflect primary igneous ratios. Also, heat producing elements, particularly U and Th, generally reside in accessory minerals such as zircon. It would be good to better characterise the samples used here. Are petrographic images or CL images available to see where the heat producing elements may be concentrated? Were the samples large or small? Do the samples show evidence for

weathering that may indicate mobilisation of U and/or Th?

Why are two different formulae to calculate heat production used here? The differences between the two methods are not explained. It is also not explicit that equations (1) and (2) refer to Methods 1 and 2.

I am not sure it is meaningful to compare these few new data to the average of the CAHFP (page 7, line 3). The CAHFP dataset includes a very large number of individual analyses that are also referenced with outcrop area. I think it is much more interesting to look at the natural range of heat production variation, rather than individual averages. Even within HPE enriched terranes, many rocks have around average heat production. This is perhaps best shown in recent work on the HPE enriched Proterozoic rocks of Australia (McLaren & Powell, JGSL, 2014). Because of the sample size issue, I am wary that there is insufficient data to dismiss the idea of elevated heat flow extending from Australia to beneath the East Antarctic ice sheet.

The author has adopted the approach of Sandiford & McLaren and Perry in estimating the vertical distribution of heat producing elements within the crust. Yet the methodology applied here relies on estimates of all parameters and is somewhat circular. Similarities in age and thickness to the Canadian and Scandinavian shields are noted but not justified with evidence in the text and need to be explained. Moreover, it is not clear to me how the measured heat production values allow a calculation of surface heat flow when other lithotypes are not considered (granites will only even be a fraction of total crustal volume). As noted on page 8 (lines 29-30) the lack of constraints on mantle heat flow, lower crustal heat production and the vertical distribution of heat producing elements contribute to very high overall uncertainties. Adopting a hr value of 7 km based on other cratonic regions doesn't take into account the potential impact of thin but highly heat producing rocks.

The potential for high heat producing rocks to be residing in the upper-middle crust (rather than on the bedrock surface permitting sampling by glacial ice) is also not ad-

dressed. The presence of these would have significant implications for sub-glacial heat flow and should be at least raised as a possibility in the Discussion section.

I think parts of the Conclusion are repetitive and unless there are specific organisational requirements for the journal, the Discussion and Conclusion sections be merged and repetition removed.

This work is important and should be further developed with additional data and more sophisticated modelling of possible/likely heat flow scenarios using additional geophysical and geological evidence. Alternatively, the current paper should include more appropriate caveats on data interpretation prior to publication.

Technical corrections

Page 3, Line 31 – change radioactive to radiogenic Figure 2 – abbreviations not explained Table 1 – it would be helpful to have the lithotype listed alongside the other sample details, also the error on the age and justification for that age being an intrusive age is required Page 7, line 20 – the model of heat production decreasing exponentially with depth is one end member Page 9, line 8 – include the value of n used for calculation of the arithmetic mean

---

## Author Comment (AC1) · 10 Nov 2017

I am grateful to the reviewer for positive comments and a statement about the value of the work. The author agrees with the reviewer that these data provide a valuable 'cross-check' for comparison with other approaches to estimating subglacial heat flow in Antarctica. As noted in separate comments, a unique aspect of the present data set is the association of rock properties with radiometric ages, thereby providing age control to the sample suite.

1. Regarding the number of samples analyzed, a similar concern was raised by Reviewer 2, and these concerns are addressed here. It would, of course, be helpful to

have a larger sample set and taken from a potentially larger catchment area. The original purpose of the clast sampling was to obtain a representative set of samples from the ice-covered East Antarctic craton in order to address questions of crustal age, composition, and history. The stated goal of the project was to investigate crustal evolution using age and isotopic constraints. That project included sampling of moraines at over a dozen sites ranging across >1,500 km of the Transantarctic Mountains (TAM). In the field, samples of any rock that represented potential Precambrian shield basement were collected. Thus, the sampling was effectively randomized. Over 300 clasts were obtained. Most of the useful yield was from only 5 sites; the others were dominated by clasts of Beacon sediment or Ferrar dolerite eroded from the TAM. A major effort was undertaken to screen the samples and set aside any with Ross Orogen ($\sim$500 Ma) ages, in order to focus solely on the Precambrian crustal history of the shield interior. This involved a significant amount of reconnaissance-type U-Pb geochronology that is normally not done as part of a petrologic study. After culling the sample suite, detailed work of mineral analysis, mineral separation, precise U-Pb geochronology, geochemical analysis, O-isotope analysis, and Hf-isotope analysis were completed over a period of several years. Work on additional samples was simply not feasible. The results of this crustal history project were published in Precambrian Research in 2017 (Goodge et al, cited). It should be emphasized that it is ONLY with high-quality age data that it is possible to then use this sample suite to constrain the interior heat-flow of East Antarctica. In the absence of age data, the origin of the clasts is completely unconstrained and a large fraction of any samples studied could well be sampling the TAM orogen or younger Beacon cover which is largely irrelevant to questions of subglacial heat flow in the craton interior. Thus, simply sampling moraines to obtain a large set of geochemical data without geochronology is likely to produce misleading results. The text in Section 2 has been revised substantially to expand upon and clarify the overall approach taken.

2. Regarding choice of samples, please see above comment and explanation added to the manuscript. To address the specific question raised here, the following revisions

have been completed: a) the names of sample sites are explicitly included in the text, differentiating the two Lonewolf Nunatak subsites A and B; b) Lonewolf Nuntaks, at the southern margin of Byrd Glacier was simply the singlemost productive site that was sampled. Presumably this is because the Byrd ice stream is among the fastest outlet glaciers traversing the TAM and capable of significant glacial erosion. It is numerically over-sampled compared to other sites, yet it contains a full range of clast ages between 1.2 and 2.0 Ga so is likely to be representative of the craton interior. c) Four sites (AGA, MRA, MSA, and TNA) were dominated by granitoids with Ross Orogen ages (18 samples) and two of these sites (AGA and MRA) yielded no pre-Ross rocks at all so are not included in this study. d) Figure 2 has been modified to show all the sites sampled, with only the three sites LWA/LWB, MSA, and TNA identified for this study. e) Manuscript text is modified to explain these points.

3. Regarding interpretation about sources in greater Wilkes Land, this is a valid point. Manuscript text is modified in section 4.3 to state that the potential sample area represents a more limited part of the Wilkes Land region.

4. Exploting ice flow. . . This is an interesting idea, but unfortunately the sample size is too small (coarse) to resolve any patterns related to individual drainages across the area. A comment to this effect was added in the Discussion section of the revised manuscript.

5. Minor revisions

a. The work of Fisher et al. (2015) does indeed represent a subglacial measurement in the area near Subglacial Lake Whillans, but as noted elsewhere in the manuscript the extremely high (anomalous) crustal heat flow value obtained is highly perturbed by advective heat transfer associated with flowing water at the base of the ice stream. As such, this measurement does not give an accurate representation of terrestrial heat flow at the base of an ice sheet. Although their result is informative of subglacial process at the Whillans Ice Stream, it does not constrain terrestrial heat flow, the subject

of this contribution. Text revised. Also note, however, a paper just published by Bege-man et al. (2017) provides a new estimate of heat flow in West Antarctica obtained by sediment probe and this work is cited in the revised manuscript.

b. Other revisions made as suggested by the reviewer.

c. Shape of catchment is not possible to address. There are published models for the extent of the Antarctic ice sheets over time (e.g., Pollard, DeConto, Scherer, etc.) and for the inheritance of subglacial drainage based on preglacial fluvial landscape (e.g., Jamieson), but to my knowledge no models are available predicting the configuration of the catchment area in which these samples were transported.

d. See above response to sample collection and sample size, including expanded explanation provided in Section 2 of text. The affect of sample size on interpretation is treated in the Discussion.

e. Two formulations to calculate heat production from geochemical compositions are commonly used, both based on rock density and concentration of the heat-producing elements U, Th and K. The approaches are both based on an original algorithm by Ry-bach (1988), modified with slightly different parameters. Both methods were included in this contribution simply for the purposes of comparison, and in order that these val-ues could more easily be compared with results from other areas that use either of the calculations. Manuscript text was revised to clarify this point.

f. Comparison with global examples. . .. Revised for clarification with specific citations. Note a new reference — recently published in 2017 — was added (Artemieva et al., 2017) on heat production in granitic rocks. Some further revisions and updates provided in the preceding paragraph as well.

g. A suggestion concerning hr. . . This is a very good idea that was not considered previously. It is a good suggestion for an independent way to assess uncertainty in this parameter. A survey of bed topography using Bedmap data across the catchment area

shows a general range of subglacial relief of 1500-2000 m, depending on how far to extend the catchment area up the flank of the Gamburtsev Subglacial Mountains. The manuscript text is revised to add this perspective.

Note: References are updated with new citations added in support of revisions and with some newly published papers of relevance (e.g., Artemieva et al., 2017; Begeman et al., 2017).
* * *
[Figure]

Heat flow (mW m$^{-2}$)

1000 km

0°

65°S

75°S

90°W

90°E

180°

Fig. 2.

**Fig. 1.** Figure 1: Terrestrial heat flow in Antarctica.

[Figure]

**Fig. 2.** Figure 2: Map showing potential source areas for dated glacial igneous clasts.

[Figure]

**Fig. 3.** Figure 3. (a) Plot of Th/U vs. K/U in glacial igneous clasts.

[Figure]

**Fig. 4.** Figure 4. Plot of surface heat production (Ho) vs. age for igneous glacial clasts.

[Figure]

Fig. 5. Figure 5. Histogram of heat flow values estimated from heat production in the glacial clasts.

[Figure]

*a* Heat flow distribution by quintile

*b* Heat flow distribution by age group

[Figure]

**Fig. 6.** Figure 6: Summary pie diagrams showing distribution of heat flow estimates from this study.

[Figure]

---

## Author Response (AR1)

**AUTHOR COMMENTS**

Title:  Crustal heat production and estimate of terrestrial heat flow in central East Antarctica, with implications for thermal input to the East Antarctic ice sheet

Author:  John W. Goodge

MS No.: tc-2017-134

(1) comments from Referees
(2) author's response
(3) author's changes in manuscript (see attached)

Reviewer comments in black type; author responses in **bold blue type**.

**Note:  References are updated with new citations added in support of revisions and with some newly published papers of relevance (e.g., Artemieva et al., 2017; Begeman et al., 2017).**

**Anonymous Referee #1**

This article written by J. Goodge is an attempt to improve our knowledge on the geothermal flux under the Antarctic Ice Sheet, and more precisely on the Byrd and Nimrod catchments. This topic is an important issue in our community, regarding the stability of the Antarctic Ice Sheet from a thermal and mechanical point of view in the future. Even if the conclusions do not change the previous estimations of the geothermal flux, the sample-based method used here constitutes a crucial cross-checking from field work. While our knowledge is mainly based on modelling results and a few analysis of temperature profiles in boreholes, this work takes advantage of glacial rock clasts gathered at the catchment output and taken as samples of the whole catchment. This original approach is valuable considering the difficulty to access the bedrock under the ice sheet. Furthermore, it allows the author to have an interesting interpretation of the heat flow distribution from a geological point of view.

The article is pleasant to read, and its structure makes the text easy to follow. Scientific context, methods, assumptions and discussion are well stated. The figures are precise and complete. The uncertainties are well evaluated and their origin discussed.

**I am grateful to the reviewer for these positive comments and statement about the value of the work. The author agrees with the reviewer that these data provide a valuable 'cross-check' for comparison with other approaches to estimating subglacial heat flow in Antarctica. As noted in separate comments, a unique aspect of the present data set is the association of rock properties with radiometric ages, thereby providing age control to the sample suite.**

General comments

Hereafter are a few remarks that could help improve the manuscript. The main concern I have is related to how the sampling process is explained.

Samples

Your results and interpretation are based on the processing of 18 samples. Of course, the processing step is heavy and it is difficult to have more samples. However, the interpretation of the distribution is based on these 18 samples only, and a different sampling could lead to significantely different interpretation. For example, only one sample correspond to a heat flow higher than 80 mw/m2, and appears like an outlier, but this could appear quite different with a larger sampling. I suggest that a small paragraph discuss this point, even if it is impossible to have more numerous samples.

**A similar concern was raised by Reviewer 2, and these concerns are addressed here. It would, of course, be helpful to have a larger sample set and taken from a potentially larger catchment area. The original purpose of the clast sampling was to obtain a representative set of samples from the ice-covered East Antarctic craton in order to address questions of crustal age, composition, and history. The stated goal of the project was to investigate crustal evolution using age and isotopic constraints. That project included sampling of moraines at over a dozen sites ranging across >1,500 km of the Transantarctic Mountains (TAM). In the field, samples of any rock that represented potential Precambrian shield basement were collected. Thus, the sampling was effectively randomized. Over 300 clasts were obtained. Most of the useful yield was from only 5 sites; the others were dominated by clasts of Beacon sediment or Ferrar dolerite eroded from the TAM. A major effort was undertaken to screen the samples and set aside any with Ross Orogen (~500 Ma) ages, in order to focus solely on the Precambrian crustal history of the shield interior. This involved a significant amount of reconnaissance-type U-Pb geochronology that is normally not done as part of a petrologic study. After culling the sample suite, detailed work of mineral analysis, mineral separation, precise U-Pb geochronology, geochemical analysis, O-isotope analysis, and Hf-isotope analysis were completed over a period of several years. Work on additional samples was simply not feasible. The results of this crustal history project were published in _Precambrian Research_ in 2017 (Goodge et al, cited). It should be emphasized that it is ONLY with high-quality age data that it is possible to then use this sample suite to constrain the interior heat-flow of East Antarctica. In the absence of age data, the origin of the clasts is completely unconstrained and a large fraction of any samples studied could well be sampling the TAM orogen or younger Beacon cover which is largely irrelevant to questions of subglacial heat flow in the craton interior. Thus, simply sampling moraines to obtain a large set of geochemical data without geochronology is likely to produce misleading results. The text in Section 2 has been revised substantially to expand upon and clarify the overall approach taken.**

More serious, I do not understand how the samples have been chosen, and why the different sampling sites are not more equally distributed. This is a problem since your conclusions are very sample-dependent. Furthermore, Table 1 is not so clear, since the acronyms LWA, LWB, MSA, TNA are never explained. I understand it refers to the sites, but I cannot figure out which one for LWB. Why Lonewolf Nunataks would be present so many times, whereas AGA and MRA disappeared? Did you sampled some rocks at AGA and MRA, and discarded them in a second step? I think §2 needs to be completed, so that your selection criteria are clearly stated.

**Please see above comment and explanation added to the manuscript. To address the specific question raised here, the following revisions have been completed:**

**a) the names of sample sites are explicitly included in the text, differentiating the two Lonewolf Nunatak subsites A and B;**

**b) Lonewolf Nuntaks, at the southern margin of Byrd Glacier was simply the singlemost productive site that was sampled. Presumably this is because the Byrd ice stream is among the fastest outlet**

**glaciers traversing the TAM and capable of significant glacial erosion. It is numerically over-sampled compared to other sites, yet it contains a full range of clast ages between 1.2 and 2.0 Ga so is likely to be representative of the craton interior.**

**c) Four sites (AGA, MRA, MSA, and TNA) were dominated by granitoids with Ross Orogen ages (18 samples) and two of these sites (AGA and MRA) yielded no pre-Ross rocks at all so are not included in this study.**

**d) Figure 2 has been modified to show all the sites sampled, with only the three sites LWA/LWB, MSA, and TNA identified for this study.**

**e) Manuscript text is modified to explain these points.**

Interpretation

Your sampling concerns the Byrd catchment mainly, which, I think, stands across Victoria Land, Adélie Land and Wilkes Land. But your interpretation concerns "the Wilkes Land region" (P.7 L.10), which is a much larger region than your source area. This is a bit confusing. Your conclusion says that an extrapolation of CAHFP under Antarctica is potentially erroneous, whereas your samples concerns a very small part of the Wilkes Land. For example, an extrapolation could be justified close to the coast of the Wilkes Land, or even at Dome C. I suggest the end of §4.3 is changed for a more careful formulation, conclusion and abstract as well.

**This is a valid point. Manuscript text is modified in section 4.3 to state that the potential sample area represents a more limited part of the Wilkes Land region.**

Exploiting the ice flow?

I was first very pleased to see that you mixed geology and ice flow aspects. However, after reading the whole paper, I felt like disappointed that you cannot fully exploit the geographical aspect given by the ice flow. The Transatarctic mountains make the problem much complex, but having a comprehensive ice-flow interpretation could be an interesting scientific perspective... to add in conclusion? In the meantime, could you at least look if there is any gradient between MSA and LWA for example, in heat flow or chemical composition? Even if it does not give any additional information, you could just mention that you looked at it.

**This is an interesting idea, but unfortunately the sample size is too small (coarse) to resolve any patterns related to individual drainages across the area. A comment to this effect was added in the Discussion section of the revised manuscript.**

Minor revisions

P2, L12 : Is the work of Fisher et al (2015) not already a direct subglacial measurement of geothermal flux ? So I would say "..., and very few from direct subglacial measurements". The same in the abstract.

**I respectfully disagree with the reviewer comment. The work of Fisher et al. (2015) does indeed represent a subglacial measurement in the area near Subglacial Lake Whillans, but as noted elsewhere in the manuscript the extremely high (anomalous) crustal heat flow value obtained is highly perturbed by advective heat transfer associated with flowing water at the base of the ice stream. As such, this**

**measurement does not give an accurate representation of terrestrial heat flow at the base of an ice sheet. Although their result is informative of subglacial process at the Whillans Ice Stream, it does not constrain terrestrial heat flow, the subject of this contribution. Text revised. Also note, however, a paper just published by Begeman et al. (2017) provides a new estimate of heat flow in West Antarctica obtained by sediment probe and this work is cited in the revised manuscript.**

P2, L24 : Since the heat budget of basal ice depends both on ice thickness and geothermal flux, there is no direct pure correlation between ice thickness and basal age, I suggest "relatively thick, slow-moving...".

**Revised as suggested.**

P3, L19 : the samples have travelled for hundred of thousands of years, and the ice sheet has not always had the present shape. In particular during glacial areas, do you have any idea of the shape of the catchment given by continental model outputs? (if possible)

**This is not possible to address. There are published models for the extent of the Antarctic ice sheets over time (e.g., Pollard, DeConto, Scherer, etc.) and for the inheritance of subglacial drainage based on preglacial fluvial landscape (e.g., Jamieson), but to my knowledge no models are available predicting the configuration of the catchment area in which these samples were transported.**

P4, L11 : how are the samples selected, and how many for each site ? Why are some sites over-represented in the number of samples? How is the choice of the sample number affect the conclusion?

**See above response to sample collection and sample size, including expanded explanation provided in Section 2 of text. The affect of sample size on interpretation is treated in the Discussion.**

P6, L21 : Could you briefly explain why it is necessary to make use of the two methods, since the results are similar? I guess that, in other circumstances, they could significantely differ?

**Two formulations to calculate heat production from geochemical compositions are commonly used, both based on rock density and concentration of the heat-producing elements U, Th and K. The approaches are both based on an original algorithm by Rybach (1988), modified with slightly different parameters. Both methods were included in this contribution simply for the purposes of comparison, and in order that these values could more easily be compared with results from other areas that use either of the calculations. Manuscript text was revised to clarify this point.**

P7, L8 : "Compared to global examples". To make it more clear, could you briefly specify, or add references?

**Revised for clarification with specific citations. Note a new reference — recently published in 2017 — was added (Artemieva et al., 2017) on heat production in granitic rocks. Some further revisions and updates provided in the preceding paragraph as well.**

P7, L19 : I think q and H are not defined, maybe change for a specification in words : "the function linking heat flow and heat production".

**Good suggestion. Revised for clarification as suggested.**

P8, L23 : A suggestion concerning hr: could you consider the variability of the bed topography on the source area as an estimator for the crust uncertainty, so that your uncertainty has a better specified origin ? I think you should be close to your present value of 1500 m and will not change your conclusions. If you disagree with this approach, could you better justify this value or add a reference?

**This is a very good idea that was not considered previously. It is a good suggestion for an independent way to assess uncertainty in this parameter. A survey of bed topography using Bedmap data across the catchment area shows a general range of subglacial relief of 1500-2000 m, depending on how far to extend the catchment area up the flank of the Gamburtsev Subglacial Mountains. The manuscript text is revised to add this perspective.**

**Anonymous Referee #2**

General comments

Goodge has developed an idea that has appeared in the literature over the past few years regarding the possible impact of high heat producing rocks on sub-glacial heat flow in Antarctica. This is an important problem given both the potentially significant impacts of sub-glacial heat flow variations on ice sheet behaviour, and the established (possibly flawed) approach of using literature averages for surface heat flow in ice sheet models. It is also an interesting area of cross-disciplinary science, and obviously a suitable topic for Cryosphere. Goodge's paper contributes to our understanding of sub- glacial heat flow variation by measuring the heat-producing-element enrichment for a limited number of assumed bedrock lithotypes sampled from moraine systems. These data are then used to estimate sub-glacial heat flow and a conclusion of average sub- glacial heat flow for East Antarctica is made. The paper is generally well-written and easy to follow but I am a bit concerned that there is insufficient new data, and caveats on the sample size and interpretation method are not adequately outlined. The sample set is very small (five localities and 18 samples) yet the conclusions are extrapolated to draw conclusions about the regional heat flow that are not necessarily justified.

**I am grateful to the reviewer for these positive comments and statement about the value of the work.**

Specific comments

There are many assumptions used in the calculations that rapidly take the reader from simple U, Th and K concentrations of a few eroded rock clasts to calculations of sub- glacial heat flow across a vast area of East Antarctica. Both a mantle heat flow, and a heat contribution from the lower crust are assumed. A length scale for heat production distribution vertically within the crust is also assumed (see further discussion below). Although I fully understand the difficulties of constraining sub-glacial geology, better constraints on mantle heat flow (from seismic velocity data for example) could be used.

**The author is not aware of any independent estimates of mantle heat flow based on seismic or other data from Antarctica. Publications using seismic velocity structure to invert heat flow (Shapiro and Ritzwoller, 2004; An et al., 2015) consider the lithosphere velocity structure and do not separately treat the mantle contribution to surface heat flow. In absence of specific constraints, an assumption of mantle heat flow based on geologically comparable craton age and thickness is a valid approach.**

The analytical approach for heat-producing element determination is fairly standard and in a geoscience paper would be included only as an appendix. I do understand that for a non-specialist geoscience audience it may be more appropriate in the main text.

**Not clear if the reviewer is recommending moving this section to an appendix. Will defer to editor recommendation.**

U and Th are known to be highly mobile elements in the near surface environment or where there is fluid flow. It would be interesting to look at the Th/U ratios (can this be added to Table 1) and the issue of mobilisation (loss or gain) of these elements to help establish that these values reflect primary igneous ratios. Also, heat producing elements, particularly U and Th, generally reside in accessory minerals such as zircon. It would be good to better characterise the samples used here. Are petrographic images or CL images available to see where the heat producing elements may be concentrated? Were the samples large or small? Do the samples show evidence for weathering that may indicate mobilisation of U and/or Th?

**It is a good suggestion to consider the potential mobility of U and Th as it relates to their contribution to heat production. As suggested by the reviewer, Table 1 has been modified to list Th/U and K/U ratios. This compilation shows that of the 18 samples presented, only one (10MSA-2.3) has anomalous element ratios. This is shown in plots of Th/U and K/U (provided as a new Figure 3), in which all of the samples show coherent behavior and can be fit to a linear regression. The ratios are also mostly within the ranges compiled in a recent paper by Artemieva et al. (2017). These relationships indicate that, in general, the samples have not experienced fluid-assisted element mobilization as reviewer has in mind. Based on the valid question posed by the reviewer, some specific comments about the element concentrations have been added to the text in Section 4.2 as well as remarks about sample characteristics and isotopic compositions to address these concerns.**

**To address the specific questions raised in this comment about sample characteristics, the reader is referred to the detailed data presented in Goodge et al. (2017), which includes sample descriptions, petrographic information, geochemical data, cathodoluminescence (CL) images of zircons, and zircon U-Pb and O stable isotope data. This is also noted explicitly in Section 2.**

Why are two different formulae to calculate heat production used here? The differences between the two methods are not explained. It is also not explicit that equations (1) and (2) refer to Methods 1 and 2.

**Please refer to author response to similar comment/question by Reviewer 1. Text revised accordingly in Section 4.3.**

I am not sure it is meaningful to compare these few new data to the average of the CAHFP (page 7, line 3). The CAHFP dataset includes a very large number of individual analyses that are also referenced with outcrop area. I think it is much more interesting to look at the natural range of heat production variation, rather than individual averages. Even within HPE enriched terranes, many rocks have around average heat production. This is perhaps best shown in recent work on the HPE enriched Proterozoic rocks of Australia (McLaren & Powell, JGSL, 2014). Because of the sample size issue, I am wary that there is insufficient data to dismiss the idea of elevated heat flow extending from Australia to beneath the East Antarctic ice sheet.

**To address a similar concern about 'few new data' that was raised by Reviewer 1, an exhaustive data set on the ages, geochemical compositions, and isotopic behavior of a large suite of igneous rocks was**

**recently published by Goodge et al. (2017, *Precambrian Research*). To my knowledge, this is the single-most comprehensive such data set on glacial clasts sampled in Antarctica to date. Unlike the case in central and South Australia, which is well studied from outcrop and industry borehole samples, it is infeasible to expand the current data set without significant additional resources to collect new samples or to conduct the various lab measurements required.**

**That said, the point raised in this part of the manuscript (original page 7, line 3) is that despite some overlap with the CAHFP, most of the glacial clast samples in this study have values of heat production less than the CAHFP average as cited. Certainly a geological province such as the CAHFP exhibits a range of heat production values, but the mean remains a valid reference point, particularly for a large data set such as noted by the reviewer. This is simply a statement of comparison and at this point does not offer any comment on possible extension of crust like that in the CAHFP into East Antarctica.**

**It is noteworthy that the paper by McLaren and Powell (2014) discusses granites from the North Australian Craton and is not directly relevant to the discussion concerning the CAHFP.**

The author has adopted the approach of Sandiford & McLaren and Perry in estimating the vertical distribution of heat producing elements within the crust. Yet the methodology applied here relies on estimates of all parameters and is somewhat circular. Similarities in age and thickness to the Canadian and Scandinavian shields are noted but not justified with evidence in the text and need to be explained. Moreover, it is not clear to me how the measured heat production values allow a calculation of surface heat flow when other lithotypes are not considered (granites will only even be a fraction of total crustal volume). As noted on page 8 (lines 29-30) the lack of constraints on mantle heat flow, lower crustal heat production and the vertical distribution of heat producing elements contribute to very high overall uncertainties. Adopting a hr value of 7 km based on other cratonic regions doesn't take into account the potential impact of thin but highly heat producing rocks.

**These are all valid criticisms offered by the reviewer, but it represents an unreasonable comparison given that the regions cited are well exposed and well studied by petrologic, geochemical, borehole, and geophysics methods. The text acknowledges (and makes explicit) the various assumptions that are required to use the available dated samples to construct a first-order profile of heat production in central East Antarctica. Likewise, uncertainties in the input parameters are explicitly considered, so it should be clear to a reader that a range in outcomes is to be expected. Certainly the crust in this region contains rock types other than just granite. We can surmise from seismology, aeromagnetics, and locally available outcrop that the East Antarctic craton is a composite of Archean to Neoproterozoic igneous and metamorphic rocks. Seismology indicates the lithosphere is thick, cold and stable. Where exposed, the rocks commonly consist of dense, dehydrated granulites, charnockites, and other gneissic rocks. Granites as a class have higher concentration of heat-producing elements than other rocks, such that a sampling of granites is likely to skew heat production (and therefore heat flow) to higher values. If anything, the data provided in this study may overestimate thermal conditions at the base of the ice sheet. To address the reviewer concerns, the text in the Introduction and Discussion have been revised to emphasize that rather than a comprehensive top-to-bottom assessment of heat production and heat flow, the data presented here provide a valuable glimpse into the thermal properties of the East Antarctic craton that at this time is otherwise inaccessible.**

The potential for high heat producing rocks to be residing in the upper-middle crust (rather than on the bedrock surface permitting sampling by glacial ice) is also not addressed. The presence of these would

have significant implications for sub-glacial heat flow and should be at least raised as a possibility in the Discussion section.

**This is a good point and was considered an implicit idea in the original manuscript, yet should be addressed explicitly. Discussion (Section 5) is revised to emphasize this point.**

I think parts of the Conclusion are repetitive and unless there are specific organisational requirements for the journal, the Discussion and Conclusion sections be merged and repetition removed.

**Author prefers to keep the Discussion and Conclusions separate, as they have different purposes. Reviewer's comment about repetition is noted and revisions have been made to both sections in order to improve the presentation.**

This work is important and should be further developed with additional data and more sophisticated modelling of possible/likely heat flow scenarios using additional geophysical and geological evidence. Alternatively, the current paper should include more appropriate caveats on data interpretation prior to publication.

**See separate comments regarding additional data. It's not clear what type of modeling is envisioned by this reviewer, but in any case modeling is beyond the scope of this contribution. The goal of this paper is to provide data relevant to ice-sheet modeling that is undertaken by others. Appropriate caveats are included in the revised manuscript.**

Technical corrections

Page 3, Line 31 – change radioactive to radiogenic

**Done.**

Figure 2 – abbreviations not explained

**Geographic abbreviations were included in caption. Explanation of sample site abbreviations has been added.**

Table 1 – it would be helpful to have the lithotype listed alongside the other sample details, also the error on the age and justification for that age being an intrusive age is required

**This information is included in Goodge et al. (2017). Age justification beyond the scope of this contribution; refer to citation. Author will provide lithotype and age uncertainty if recommended by the editor, but this will expand Table 1.**

Page 7, line 20 – the model of heat production decreasing exponentially with depth is one end member

**Absolutely, but it is a commonly assumed model based on a paucity of observations from exposed crustal sections.**

Page 9, line 8 – include the value of n used for calculation of the arithmetic mean.

**Done.**

[revised manuscript text omitted]

---

## Author Response (AR2)

**tc-2017-134**

Dear Olaf,

Thanks for your positive decision on my manuscript. I am currently in Antarctica finishing up my field project and have been able to access the author information. I am submitting a revised manuscript with all of the changes you recommend to the text, Table 1, and Fig. 2. These are highlighted in blue text.

I appreciate your prompt attention to my manuscript in November. Unfortunately, your reply was not forwarded to me while you were traveling and before I left for the ice, but I am happy to have your feedback now and to make the suggested changes.

I should be home by Jan 12, after which I will be available for any questions or concerns about copy-editing.

Again, thank you for your handling of this manuscript. It's my first time working with The Cryosphere as an author and I appreciate your openness to this cross-disciplinary contribution.

Cheers,

John